# MiloNet: A Framework for Traceable and Verified RAG

## Abstract

Retrieval-augmented generation (RAG) has improved factual grounding, but strong retrieval alone does not guarantee reliable answers. Systems can still produce unsupported claims or omit required information after relevant evidence has been found. We introduce MiloNet, a framework for traceable RAG in text-grounded settings that combines summary-based routing, hierarchical evidence construction, cited synthesis, and final answer checks to control which evidence is admitted and which content can appear in the final answer. Under our evaluation protocol on the RAGBench HotpotQA test split, MiloNet-full achieves near-zero unsupported-claim rates, reduces omissions, and improves faithfulness and answer quality over the evaluated baselines. These results suggest that, in the evaluated RAGBench HotpotQA setting, hierarchical evidence construction, cited synthesis, and final answer checks help reduce unsupported claims and omissions after relevant evidence has been retrieved.

## 1 Introduction

Retrieval-augmented generation (RAG) gives language models access to external context Lewis et al. (2021) and is now a standard approach for knowledge-intensive tasks. A basic RAG pipeline retrieves evidence once and then generates an answer from the retrieved context. Recent work has improved many parts of this pipeline. This includes better retrieval representations, such as HypRAG Madhu et al. (2026), adaptive retrieval and self-critique, such as Self-RAG Asai et al. (2023), and hierarchical summarisation for long-document retrieval, such as RAPTOR Sarthi et al. (2024). Benchmarks such as RAGBench Friel et al. (2025) have also made it easier to evaluate retrieval quality and answer quality separately.

Current RAG research is moving in several connected directions. Retrieval is increasingly treated as a multi-step process rather than a single action. For example, CoRAG retrieves and reasons in steps with explicit test-time scaling Wang et al. (2025), while CoopRAG combines LLM-based questioning with uncertainty-aware reasoning and reranking Ko et al. (2025). Retrieval is also becoming more structured. GFM-RAG, HiRAG, and MixRAG introduce graph or hierarchical structure into indexing and retrieval to better capture multi-hop relationships in the evidence corpus Luo et al. (2025); Huang et al. (2025); Liu et al. (2026). A further line of work focuses on efficiency under realistic serving constraints. PCED moves evidence fusion from long-prompt attention into decoding Corallo & Papotti (2026), and deployment studies show that retrieval gains can shrink once reranking, truncation, and latency limits are enforced Medrano et al. (2026).

These advances improve retrieval, representation, and efficiency. They do not by themselves solve a central reliability problem. Retrieved context can still hurt answer quality when it is on-topic but misleading, noisy, or overly heterogeneous. This has motivated work on source reliability and robustness, including RA-RAG and RAGUARD Hwang et al. (2025); Zeng et al. (2026). In practice, many failures arise after evidence has already been found. The remaining question is whether that evidence is traceable, admissible, and carried into the final response without unsupported additions or missing required content.

We address this problem with MiloNet, a framework for traceable RAG in text-grounded settings where answers should remain tied to a fixed textual evidence corpus. MiloNet aims to reduce unsupported claims and omissions at inference time through three design choices. First, summary-based semantic routing over document-level summaries builds a bounded allowlist of eligible document tools before fine-grained retrieval begins. Second, retrieval and evidence construction are organised as a hierarchical con-

troller–folder–document agent system. Third, final answer checks and admissibility-based response composition filter unsupported or out-of-scope content before the final answer is emitted. In this sense, MiloNet acts as an answer-governance architecture for RAG. It complements recent work on multi-step retrieval Wang et al. (2025), structured retrieval Luo et al. (2025), source reliability estimation Hwang et al. (2025), and decoding-time efficiency Corallo & Papotti (2026). The paper also introduces an evaluation protocol that separates evidence coverage from grounded synthesis on the RAGBench HotpotQA split.

## 2   Related Work

Broadly, recent work in RAG has developed along several connected lines. The first is improved retrieval under limited context budgets, the second focuses on selecting the most useful evidence from a candidate set, and the third is grounding through verification. Alongside these, benchmarks and evaluation frameworks have made evidence attribution easier to measure Friel et al. (2025); Ren et al. (2026); Es et al. (2024).

A representative retrieval-oriented approach relevant to the present work is Recursive Abstractive Processing for Tree-Organized Retrieval (RAPTOR), which builds a multi-level tree where leaf nodes represent base text units and internal nodes store abstractive summaries of their descendants Sarthi et al. (2024). This supports retrieval over multiple abstraction levels, constrained by a node budget. Although RAPTOR was originally designed for long-form documents, the same structure can be adapted to collections of shorter documents by treating each document as a leaf-level unit. In our experiments, we use a similar document-level variant to avoid introducing an additional chunking stage, as described in §4.2. In essence, hierarchical retrievers improve evidence organisation and context assembly.

Complementary to this is work using LLM-based reranking to choose and order candidate documents before generation. RankGPT uses listwise ranking with large language models and is a strong baseline when candidate recall is already high and the main challenge is selecting the right evidence Sun et al. (2024). In our implementation, we instantiate a RankGPT-like process as a joint selection-and-ranking baseline so that it can both filter and order candidate documents. While this is valuable, stronger selection alone does not prevent unsupported details from entering the final response or guarantee completeness with respect to the available evidence.

Verification-oriented generation addresses this problem. Approaches such as Self-RAG introduce a model-centric direction in which the model learns to retrieve on demand and critique its own generations using special tokens and feedback-style signals Asai et al. (2023). Rather than taking this approach, in the present work we focus on inference-time reliability within a modular retrieval pipeline, without task-specific fine-tuning. Recent benchmarks such as SIN-Bench are also relevant here, as they extend evaluation toward verification in long-context multimodal settings Ren et al. (2026). For benchmarking in the present work, we focus on textual claim alignment and use the sentence-level annotations in RAGBench to measure hallucination and omission directly.

In summary, our work is closest to the reliability-oriented research focus, but differs in two ways. First, it treats reliability as an inference-time framework problem rather than as a retrieval, reranking, or model-training problem. Second, it separates evidence coverage from synthesis quality. MiloNet does this through corpus-scale routing, hierarchical candidate construction, citation-bound synthesis, and verification-heavy postprocessing.

## 3   Methodology

We present **MiloNet**, a hierarchical RAG system that uses bounded routing to control the candidate evidence space in a three-stage inference pipeline. First, preprocessing performs corpus-scale routing over document summaries. Second, a multi-layer hierarchical core RAG structure, comprising SubCoA, folder, and document agents, constructs grounded candidate answers, where 'SubCoA' denotes the top-level Chain-of-Abstraction (CoA)-style agent Gao et al. (2025) that coordinates hierarchical evidence construction and produces the aggregated output used for postprocessing. Third, verification-focused postprocessing suppresses unsupported claims before response composition (Fig. 1–3). This design supports routing over the

pooled corpus of $N$ document-level agents while constraining response composition to admissible candidate units with document-ID citations. The system explicitly flags any supporting statement that lacks a valid citation. The trade-off is that systematically reducing false-positive and false-negative claims in this way may come with increased inference costs, because of the additional verification calls needed.

For a query $q$, preprocessing constructs a candidate pool $C(q)$ and a committed tool-filter allowlist $A(q)$ of document-level tool IDs, while postprocessing admits a candidate unit $u$ into final response composition only if the admissibility predicate $\mathrm{Adm}(u)$ holds. The complete notation is summarised in Appendix A.1.

### 3.1 Preprocessing with Summary-Based Semantic Routing

To enable corpus-scale routing over the evaluated 1,558-document pooled corpus, MiloNet performs summary-based semantic routing during preprocessing. Instead of indexing full document texts at the routing stage, it indexes dense embeddings of document summaries in a shared ChromaDB vector store. This reduces topical noise during routing and defers fine-grained evidence gathering to downstream agents.

In this setting, document-level tool IDs refer specifically to leaf tools that invoke document agents. Summary-based routing represents each source with a fixed-size summary embedding and filters irrelevant sources early, before document-local retrieval and verification. In our current instantiation, routing nodes correspond to document-level summaries.

As shown in Fig. 1, MiloNet builds the downstream tool-filter allowlist via multiple ID sources over a shared summary index. First, a summary-index query over the original user input seeds the ordered candidate pool $C(q)$ with document-level tool IDs. In addition, a CoA-style planner and a reasoning-and-acting (ReAct)-style planner Yao et al. (2023) invoke the same tracked retriever over the summary index during planning, while the corresponding tracked IDs remain buffered until the end of preprocessing.

MiloNet also uses the two planner outputs to select a textual draft for canonicalisation. This textual selection is independent of tracked tool-call buffering, which remains separate until the final tracker flush.

The canonicalised draft may then trigger an auxiliary summary lookup (i.e., an optional second summary-index retrieval pass) to enrich the candidate pool before downstream execution. Query refinement is applied after this second pass. When the canonicalised draft is sufficiently informative, MiloNet may append salient focus terms to produce a refined query for the subsequent core RAG stage.

At the end of preprocessing, planner-tracked IDs are flushed once under a small admission cap and merged with metadata-seeded IDs from summary retrieval passes. The merged pool is then de-duplicated in order-preserving fashion and truncated to the global capacity to form the final allowlist. MiloNet intentionally avoids a single global re-ranking over all candidates because these candidates arise from heterogeneous signals that are not always directly comparable.

At this stage, preprocessing produces only routing constraints and an optional refined query, not user-facing factual claims. Further implementation details are provided in Appendix A.2.

### 3.2 Core RAG with Hierarchical SubCoA–Folder–Document Agents

After preprocessing produces the tool-filter allowlist $A(q)$ of eligible document-level tool IDs, MiloNet performs core retrieval and evidence construction through a hierarchical SubCoA-folder-document structure, as shown in Fig. 2. SubCoA and folder agents act as controllers for planning and aggregation, whereas document agents are leaf-level tools that execute document-local retrieval and candidate extraction. At runtime, allowed document-level tool IDs are mapped to their parent folder agents so that downstream tool exposure is restricted accordingly.

Across SubCoA, folder, and document agents, MiloNet uses role-specific instruction templates to constrain outputs into verifiable intermediate representations. This reduces free-form generation during evidence construction and bridges corpus-scale routing with document-local evidence gathering.

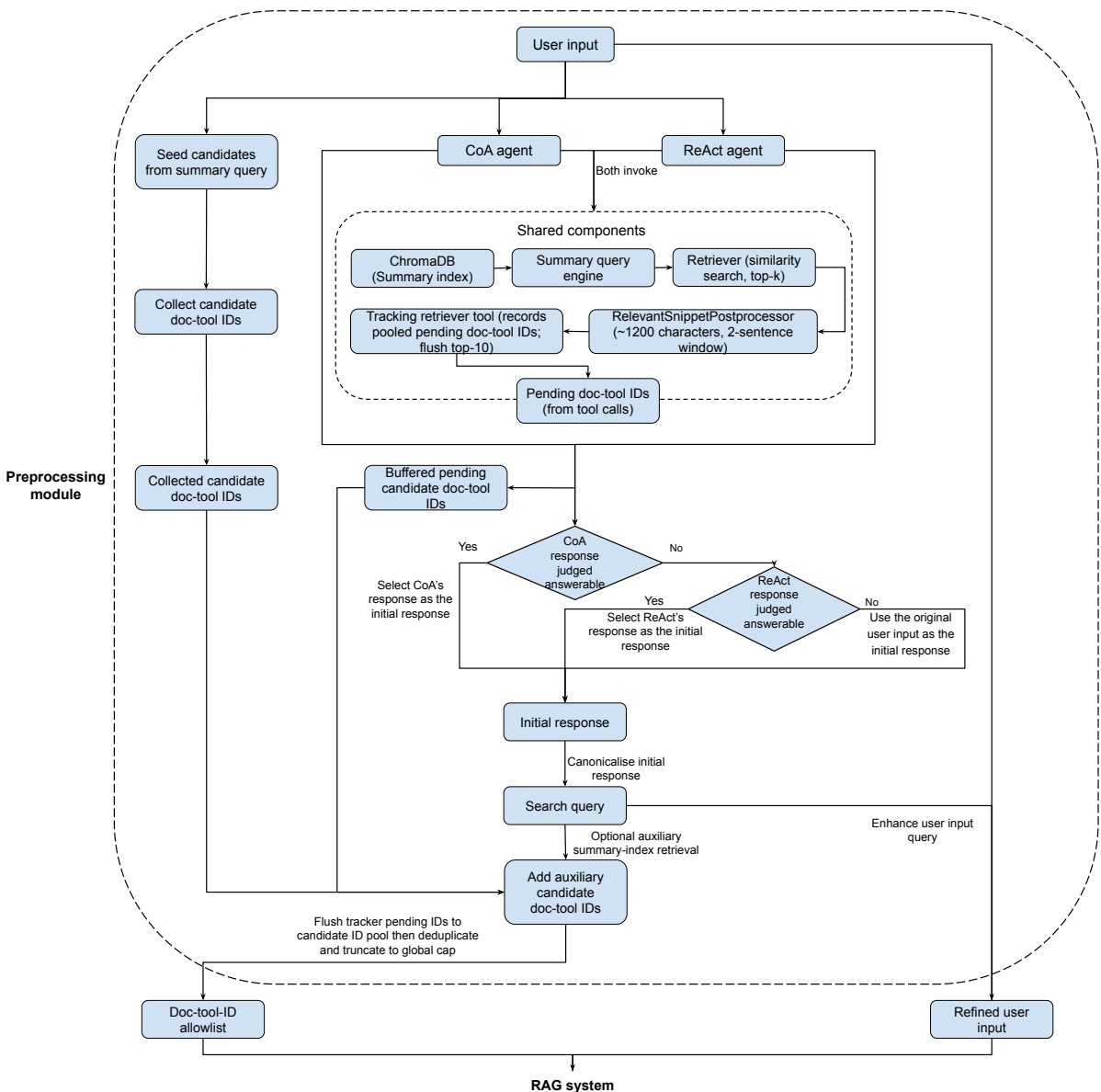

Figure 1: MiloNet preprocessing uses a two-stage summary-index retrieval process together with tool-call tracking from the CoA and ReAct planners to construct a document-level tool-ID allowlist and a refined query for downstream core RAG. We denote the candidate ID pool as $C(q)$ and the committed allowlist as $A(q)$.

**Reference disambiguation for verified facts**  Before promoting folder-level outputs to higher-level composition, MiloNet applies a lightweight reference disambiguation step to each verified fact sentence using the originating document context. This step is faithfulness-preserving and reduces cross-sentence ambiguity during multi-source aggregation.

**Parallel evidence construction**  For a given query, MiloNet follows a hierarchical fan-out–fan-in execution pattern. A SubCoA agent sends the same wrapped query string to multiple folder agents, which run independently and can be executed in parallel. Each folder agent then sends the same query string to its

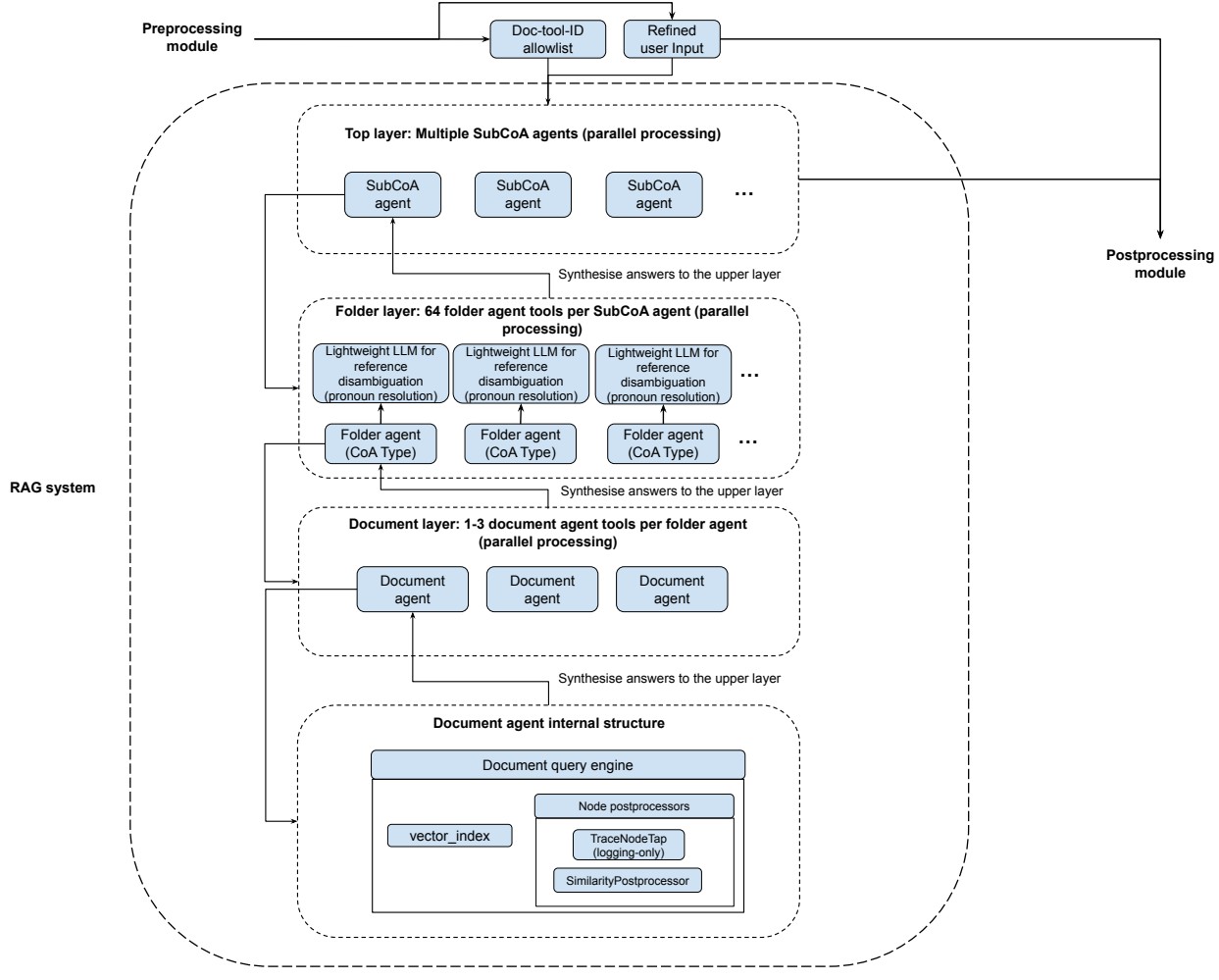

Figure 2: MiloNet core structured RAG with hierarchical SubCoA-folder-document agents for candidate unit construction with controlled promotion of sources.

document agents in parallel and aggregates the returned candidate units. The wrapped query string is not rewritten per agent during this stage, so differences in retrieved evidence arise from agent-specific context and document-local indices rather than from per-agent query reformulation.

At the document level, each document agent combines intra-document retrieval with a local postprocessing pipeline that converts retrieved snippets into structured candidate units. A candidate unit is an intermediate fact representation paired with document provenance. It is treated as a proposal rather than a trusted fact and must pass downstream verification before final response composition.

Overall, the core RAG stage transforms the preprocessing-derived allowlist into document-cited candidate units for downstream postprocessing. Further implementation details are provided in Appendix A.3.

### 3.3 Postprocessing with Verification-Driven Filtering and Response Composition

As shown in Fig. 3, postprocessing governs how MiloNet admits candidate units into the cited response. We write $\mathrm{Adm}(u)$ for the admissibility predicate induced by postprocessing. A unit is eligible for composition only if it carries an explicit document identifier and satisfies the downstream postprocessing constraints.

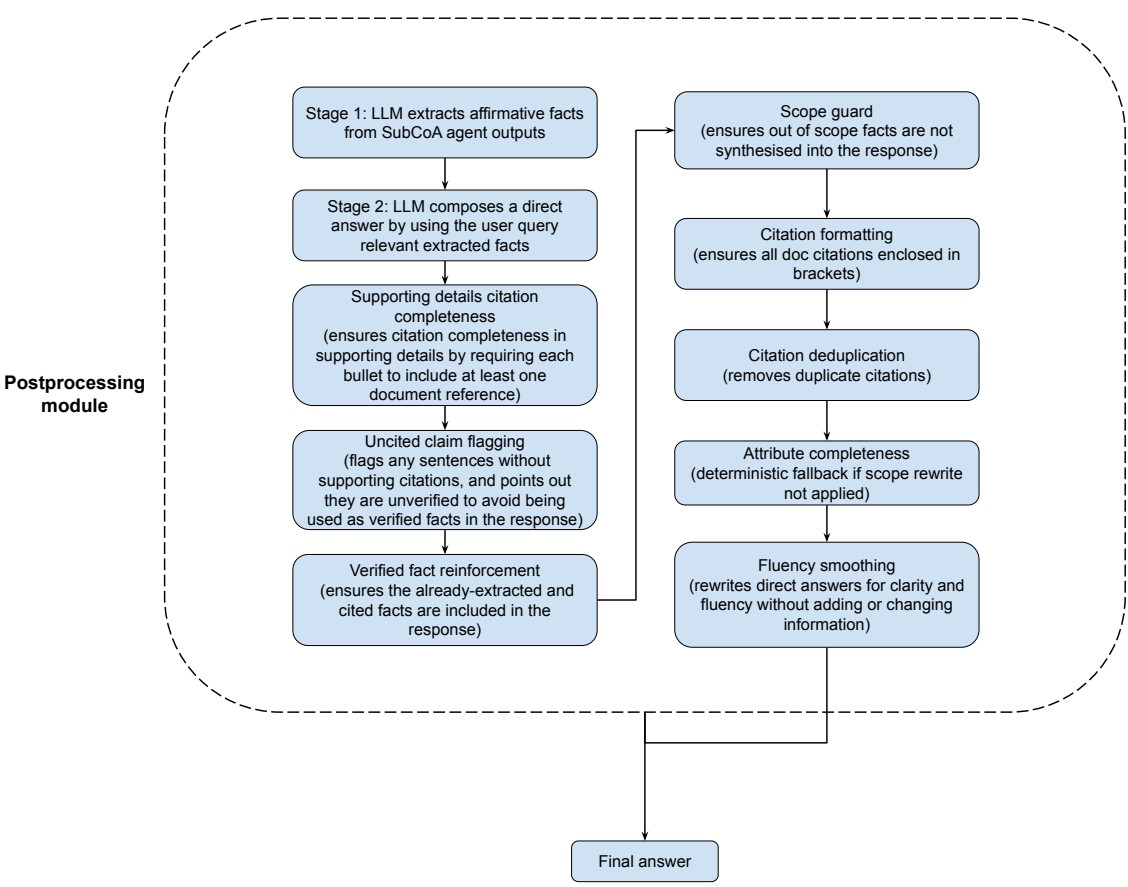

Figure 3: MiloNet postprocessing consolidates candidate units, produces a cited answer with the Stage 2 synthesis prompt, checks citation completeness and scope, formats citations, and suppresses false positives before emitting the final answer. The comparison between MiloNet-core and MiloNet-full in §5.4 holds candidate consolidation and verified-fact extraction fixed, but the settings use different synthesis prompts. MiloNet-core uses the verified-fact synthesis prompt and stops after that synthesis call, whereas MiloNet-full uses the constraint-guided synthesis prompt and continues with the subsequent final answer checks. The admissibility predicate Adm($u$) determines which units enter response composition.

**Stage 1 (candidate consolidation and citation preparation)**  The system consolidates candidate units from multiple SubCoA agents, normalises their representations, and associates each unit with explicit document-ID citations. Optional lightweight ambiguity reduction may also be applied while preserving faithfulness.

**Stage 2 (cited answer synthesis)**  Both MiloNet-core and MiloNet-full perform Stage 2 synthesis over the extracted, document-cited facts, but use different synthesis prompts. MiloNet-core stops after its synthesis call, whereas MiloNet-full proceeds to the subsequent final answer checks.

**Admissibility and neutral handling of missing evidence**  MiloNet requires each intermediate claim in a multi-step conclusion to be individually supported by cited candidate units. When sufficient support is unavailable, the system suppresses the unsupported conclusion and returns either a partial answer containing only document-cited statements or an explicit statement of insufficient evidence.

These postprocessing checks include citation completeness, uncited-statement flagging, scope and attribute validation against supporting details, citation formatting normalisation, and fluency-only cleanup without

introducing new factual content. Further details on the two synthesis prompts and the subsequent final answer checks are provided in Appendix A.4.

In summary, postprocessing converts open-ended generation into constrained composition built from admissible, document-cited candidate units, with any uncited statements explicitly flagged.

# 4 Experimental Setup

## 4.1 Dataset and corpus construction

We evaluate on the RAGBench HotpotQA test split.[1] For each query $q$, the benchmark provides a query-specific candidate set of documents $D_q$ (typically 2–4 documents; $|D_q| = 4$ for 388 out of 390 queries) as plain text together with sentence-level annotation keys (e.g., `0a`, `1c`) used by the benchmark.[2] We construct a pooled corpus by collecting the unique documents appearing across this split, creating a document-agent corpus of size $N$ (in our experiments $N$=1558), following the notation in Appendix A.1.

Most baselines are evaluated in the standard RAGBench setting where retrieval operates over the per-query candidate set $D_q$. In contrast, MiloNet performs corpus-scale routing over the pooled corpus and must retrieve relevant documents without access to $D_q$. All reported comparisons are computed on the test split (390 queries) for every system.

## 4.2 Systems compared

We compare MiloNet against representative RAG baselines spanning direct retrieval, reranking, and hierarchical indexing. Table 1 summarises the key architectural differences and the retrieval setting for each system. We interpret these comparisons as system-level comparisons under a shared sentence-key scoring protocol with deterministic system-specific adapters, not as a pure backbone-isolated leaderboard. The controlled internal comparison is between MiloNet-core and MiloNet-full. Both variants share the same upstream evidence pipeline, including summary-index routing, bounded evidence admission, the SubCoA–folder–document hierarchy, and verified-fact extraction. Both variants then perform Stage 2 cited answer synthesis using different prompts. MiloNet-core stops after its synthesis call, whereas MiloNet-full continues with the final answer checks.

**MiloNet variants** **(i) MiloNet-full** runs the complete three-stage pipeline. It includes preprocessing for summary-index routing and allowlist construction, hierarchical core RAG (SubCoA-folder-document agents) for candidate-unit construction, verified-fact extraction, and the complete answer-finalisation stage. This stage consists of constraint-guided cited synthesis followed by final answer checks for support, scope, citation formatting, and attribute consistency.

**(ii) MiloNet-core** uses the same preprocessing, the same core RAG stage, and verified-fact extraction. It then uses the verified-fact synthesis prompt and stops after the Stage 2 synthesis call, without running the subsequent final answer checks.

**Baselines** **(iii) Vanilla RAG** performs lexical sentence retrieval on $D_q$. Sentences are ranked by normalised token overlap with the query, and the top-$k$ are selected as context. To ensure a fair comparison under an evidence-count aligned setting, $k$ is set per-query to match the number of context blocks used by MiloNet on the same question (*context-equalised* selection).

**(iv) RankGPT RAG** implements a reranking-based pipeline aligned with the RankGPT approach. Given the benchmark-provided documents, it uses an LLM reranker to order candidates and then generates from the top-ranked context. Concretely, the ranker is prompted to *select only* the documents that provide the information needed to answer the question and to return their indices in descending relevance (e.g., `[0,2]`), yielding an *adaptive* selection size rather than a fixed top-$k$ truncation. When $|D_q| \leq 1$, RankGPT selects the single available document. If the ranker returns an empty selection, it falls back to selecting the first

---

[1]Dataset viewer: `galileo-ai/ragbench`, `hotpotqa/test`.
[2]RAGBench cases provide per-query documents without persistent global document identifiers.

Table 1: System architecture comparison. $D_q$: per-query candidate set, and $|D_q| = 4$ for nearly all queries (388 out of 390); $\mathcal{D}$: pooled corpus ($N$=1558). The RAGBench reference baseline uses the benchmark-provided response and gold sentence keys, with no retrieval or routing, under the same sentence-key scoring protocol. We report the RAGBench column as a reference condition.

| | MiloNet-core | MiloNet-full | Vanilla RAG | RAPTOR | RankGPT | RAGBench reference baseline |
|---|---|---|---|---|---|---|
| **Doc pool for retrieval** | $\mathcal{D}$ (corpus) | $\mathcal{D}$ (corpus) | $D_q$ | $D_q$ | $D_q$ | $D_q$ |
| **Document isolation** | ✓(tool per doc) | ✓(tool per doc) | × | × | × | × |
| **Structure** | SubCoA-folder-doc agents | SubCoA-folder-doc agents | Flat sentence selection | Per-case micro-tree | Flat list of documents | Flat list of documents |
| **Routing** | Summary-index allowlist $A(q)$ | Summary-index allowlist $A(q)$ | Lexical overlap top-$k$ sentences | Cluster $\rightarrow$ flatten $\rightarrow$ rank | LLM selects+ ranks docs | Benchmark-provided evidence |
| **Dual-gate planner** | ✓(CoA/ReAct gate) | ✓(CoA/ReAct gate) | × | × | × | × |
| **Verified facts / cited synthesis** | ✓ | ✓ | × | × | × | × |
| **Final answer checks** | × | ✓ | × | × | × | × |
| **Response composition** | Cited units + verified-fact synthesis | Cited units + constraint-guided synthesis | Selected sentences from $D_q$ | Parent summaries + doc summaries + raw $D_q$ | Selected docs from $D_q$ | Benchmark-provided evidence sentences |
| **LLM (our runs / provided)** | gpt-4.1-mini (+o4-mini/gpt-5-nano) | gpt-4.1-mini (+o4-mini/gpt-5-nano) | o4-mini | o4-mini | o4-mini | gpt-3.5-turbo |

document. We keep RankGPT in this intended reranking setting (i.e., operating on a high-recall candidate set) to avoid conflating reranking quality with upstream retrieval.

**(v) RAPTOR (c4_k12)** implements hierarchical indexing via summary embeddings and per-case structure induction. For each query, RAPTOR starts from the benchmark-provided document set $D_q$ and retrieves summaries for its documents from a pre-computed global summary store using document-text-to-summary embedding similarity rather than hard document-ID lookup. This design is aligned with RAPTOR's original use of semantic structure in embedding space Sarthi et al. (2024). In our implementation, the retrieved summaries are then clustered into a per-case micro-tree, flattened into a unified node pool (cluster summaries, document summaries, and raw document text from $D_q$), ranked by embedding similarity to the query, and the top-$k$ nodes are selected for context assembly to generate the answer.

We evaluated a set of $(c, k)$ configurations, where $c$ is the number of selected clusters and $k$ is the global top-$k$ node budget, using the same evaluation pipeline described in § 4.3. We selected `c4_k12` because it achieved low FP and FN rates while keeping the retrieval budget moderate among the tested configurations. The full setup and selection rationale are given in Appendix B. Under `c4_k12`, the ranked node pool includes raw-document nodes from $D_q$ in addition to summaries, so RAPTOR can still surface gold relevant sentence keys even when summary nodes are imperfect matches.

**(vi) RAGBench reference baseline** uses the benchmark-provided gpt-3.5-turbo response for answer-level evaluation and the benchmark-provided gold relevant and utilised sentence keys for alignment metrics. It performs no retrieval, routing, or adapter mapping and is evaluated under the shared sentence-key scoring protocol.

## 4.3 Evaluation protocol

All systems are evaluated using the same downstream evaluation protocol. For each query, we (i) record the system's retrieved contexts and final response, (ii) map the system's native retrieved context identifiers

| System | Native retrieved context identifiers | Mapped to sentence keys |
|---|---|---|
| MiloNet | unique document identifiers | match benchmark sentences in retrieved document context blocks |
| Vanilla RAG | sentence keys | sentence keys recorded directly |
| RankGPT | query-local document indices | expand selected documents to sentence keys |
| RAPTOR | selected nodes with document indices | expand selected documents to sentence keys |

Table 2: Adapter layer mapping heterogeneous system outputs to RAGBench sentence keys for unified evaluation.

to RAGBench sentence keys, (iii) score responses using a shared response-quality evaluation procedure, and (iv) run a diagnostic process that attributes errors to diagnostic false positives (DFPs) and diagnostic false negatives (DFNs), producing per-system summaries over all 390 cases.

**Unified sentence-key identifiers**   We compute retrieval and generation metrics in the sentence-key space provided by RAGBench (§4.1) by mapping each system's native retrieved context identifiers to sentence keys via a lightweight adapter layer. To enable unified evaluation across architecturally heterogeneous systems, we implement an adapter layer that maps each system's native retrieval output to this common sentence-key space, whether the output is expressed as document IDs in MiloNet or, in baselines, as sentence keys, query-local document indices, or selected nodes with document indices. For MiloNet, sentence keys are obtained by matching benchmark sentences appearing in the retrieved document context blocks. All set-based retrieval-level and generation-level metrics are computed on sentence keys after this deterministic mapping. Table 2 summarises the native reference format of each system and its mapping to sentence keys.

For RAPTOR, retrieval is performed over a flattened node pool derived from a per-case hierarchy, rather than directly over benchmark sentence keys. We map its selected nodes back to the underlying benchmark documents (via the referenced document indices) and expand those documents to sentence keys for set-based alignment metrics, since benchmark sentence keys are defined only over the original documents.

**DFPs and DFNs**   The diagnostic process is executed on the query, the final response, and the benchmark documents. It flags unsupported claims as DFPs, and it also flags incorrect denials when the response asserts that information is absent even though it is contained in the benchmark documents. It flags missing required information as DFNs when the benchmark documents contain the necessary answer but the response fails to provide it or provides an incorrect or misleading answer. We report case-level diagnostic false-positive rates (DFPR) and diagnostic false-negative rates (DFNR), defined as the fraction of queries with at least one diagnostic false-positive flag and at least one diagnostic false-negative flag, respectively.

## 4.4   Metrics

We report a multi-layer metric suite that separates alignment metrics, diagnostic error rates, and rubric-based quality ratings. Our primary objective is to minimise omissions and unsupported claims (hallucinations), which are reflected by DFNR, DFPR, and faithfulness. Alignment metrics ($rr$, $rp$, $gr$, $gp$) are computed in the adapter-mapped sentence-key space to enable consistent set-based comparison across systems. We interpret precision-based metrics ($rp$, $gp$) with caution because gold relevant and utilised keys are sparse, and precision is sensitive to how retrieved and used keys are extracted and mapped across systems (Appendix D and Appendix E). We therefore report DFPR, DFNR, mean answer quality, answer-quality failure rate, faithfulness $f$, and recall-based alignment metrics ($rr$, $gr$) in the main results table, and include $rp$, $gp$, F1-based metrics ($r\_f1$, $g\_f1$), trustworthiness $t$, and correctness as supplementary results in Appendix C. The answer-quality failure rate is the fraction of responses with rubric score below 3.

**Alignment metrics**   All alignment metrics are computed in the adapter-mapped sentence-key space. Let $G_q$ denote the gold relevant sentence keys and let $R_q$ denote the retrieved sentence keys after adapter

mapping. We define retrieval recall as

$$rr = \frac{|R_q \cap G_q|}{|G_q|},$$

with the convention that the ratio is zero when the denominator is empty.

Let $U_q$ denote the gold utilised sentence keys and let $S_q$ denote the sentence keys used by the system in producing its response. We define generation recall as

$$gr = \frac{|S_q \cap U_q|}{|U_q|},$$

with the same convention for empty denominators.

**Faithfulness**  We compute a faithfulness score $f$ using the RAGAS framework Es et al. (2024). Given a response, the evaluator first decomposes it into a set of atomic claims $\mathcal{C}$. Each claim $c \in \mathcal{C}$ is then checked against the provided documents using an LLM-based claim support evaluator. Faithfulness is defined as the proportion of claims that are supported.

$$f = \frac{|\{c \in \mathcal{C} : \text{supported}(c, D_q)\}|}{|\mathcal{C}|}.$$

In our evaluation, $D_q$ denotes the benchmark-provided query-specific candidate set of documents. This evaluates claim grounding against the benchmark candidate set $D_q$, rather than internal consistency with the system-retrieved contexts.

**Diagnostics and quality rubric**  We report DFPR and DFNR from the diagnostic process described in §4.3. We also report a rubric-based quality score for each response and answer-quality failure rate, which measures the fraction of responses that do not meet the highest quality standard.

We treat rubric scores as an overall answer-level quality indicator, and diagnostics as a claim-level audit that is sensitive to localised FP and FN errors. These signals are complementary and need not coincide on every instance.

Quality scores are assigned by an LLM evaluator on a 0–3 integer scale based on alignment with the per-query candidate documents ($D_q$) and overall answer quality. All judgements of support, contradiction, and missing information are made with respect to $D_q$.

A score of 0 indicates an entirely incorrect response with respect to $D_q$, which may contradict $D_q$, fabricate key facts, or incorrectly state that information is not present when it is supported in $D_q$.

A score of 1 indicates a mixed or incomplete response that contains both correct and incorrect content, and may include errors, unsupported claims, or significant omissions of document-supported information needed to answer the query.

A score of 2 indicates a mostly correct response with minor issues where all major claims are supported by $D_q$, but the response may contain small inaccuracies, incomplete coverage, or minor unsupported inferences. It provides genuine value and correctly identifies what can be determined from $D_q$.

A score of 3 indicates an excellent response that is accurate and complete with respect to $D_q$, clearly explains reasoning when appropriate, acknowledges limitations, and adds value without material hallucination. This category also includes responses that correctly state when the requested information cannot be determined while still providing all available relevant information from $D_q$. Let $s(q)$ denote the quality score for query $q$. The mean answer quality reported in Table 3 is

$$\frac{1}{|\mathcal{Q}|} \sum_{q \in \mathcal{Q}} s(q).$$

We define answer-quality failure rate as

$$\text{AQFR} = \frac{1}{|\mathcal{Q}|} \sum_{q \in \mathcal{Q}} \mathbf{1}\big[s(q) < 3\big],$$

| System | DFPR ↓ | DFNR ↓ | Mean answer quality ↑ | Answer-quality failure rate ↓ | $f$ ↑ | $rr$ ↑ | $gr$ ↑ |
|---|---|---|---|---|---|---|---|
| MiloNet-full | **0.005** | **0.003** | **2.997** | **0.003** | **0.938** | 0.994 | 0.963 |
| MiloNet-core | 0.018 | 0.038 | 2.923 | 0.038 | 0.850 | 0.994 | 0.928 |
| Vanilla RAG | 0.369 | 0.362 | 1.918 | 0.390 | 0.748 | 0.653 | 0.663 |
| RankGPT | 0.097 | 0.095 | 2.749 | 0.100 | 0.858 | 0.909 | 0.915 |
| RAPTOR c4_k12 | 0.100 | 0.087 | 2.733 | 0.097 | 0.855 | **1.000** | 0.981 |
| RAGBench reference baseline | 0.126 | 0.115 | 2.659 | 0.138 | 0.840 | **1.000** | **0.982**[†] |

Table 3: Main results on the HotpotQA test split (390 queries). DFPR and DFNR are case-level diagnostic error rates for unsupported claims and omissions. Mean answer quality is computed on the 0–3 rubric, and answer-quality failure rate is the fraction of responses with rubric score below 3. We also report faithfulness $f$ and recall-based alignment metrics $rr$ and $gr$. Higher is better for mean answer quality, $f$, $rr$, and $gr$, while lower is better for DFPR, DFNR, and answer-quality failure rate. RAPTOR uses `c4_k12`, selected via the sweep in Appendix B. The RAGBench reference baseline uses the benchmark-provided response and gold sentence keys under the same sentence-key scoring protocol. [†]The reference baseline has $gr < 1.0$ because RAGBench contains 7 instances with an empty gold utilised set ($|U_q| = 0$); following our convention, empty-denominator ratios are set to zero.

where $\mathcal{Q}$ is the set of evaluation queries and $\mathbf{1}[\cdot]$ is the indicator function.

# 5 Results and Analysis

We evaluate all systems on the 390-query RAGBench HotpotQA test split under the protocol in §4.3. Unless stated otherwise, faithfulness and diagnostics are computed against the benchmark candidate documents $D_q$. Our analysis prioritises controlling unsupported claims (the 0-FP goal) and also measures omissions, as reflected by DFPR, DFNR, faithfulness, mean answer quality, and answer-quality failure rate.

## 5.1 Overall reliability and low-quality cases

Table 3 summarises end-to-end results. MiloNet-full achieves the lowest diagnostic error rates, with 2 issue cases out of 390 and DFPR of 0.005 and DFNR of 0.003. By contrast, MiloNet-core produces 16 issue cases, indicating that the complete answer-finalisation stage further reduces both unsupported claims and omissions. Among baselines, RankGPT and RAPTOR exhibit similar diagnostic error rates, whereas Vanilla RAG shows substantially higher error rates.

The answer-quality columns show the same pattern under the 0–3 rubric. MiloNet-full reaches a mean answer quality of 2.997 and an answer-quality failure rate of 0.003. MiloNet-core also remains high at 2.923, while the baselines produce lower mean scores and a larger fraction of responses below 3.

## 5.2 Faithfulness and reference baseline

Faithfulness directly measures claim support against the benchmark-provided candidate documents $D_q$. MiloNet-full achieves the highest faithfulness (0.938), improving over the strongest baselines among the non-reference systems (RankGPT: 0.858; RAPTOR: 0.855). This aligns with MiloNet's design that constrains synthesis to admissible units with document-grounded citations and applies final answer checks for support, scope, citation formatting, and attribute consistency.

Composite scores such as trustworthiness and correctness are reported only as supplementary diagnostics in Appendix C. They are not used as primary evidence in the main comparison.

The RAGBench reference baseline uses the benchmark-provided response and gold sentence keys under the same scoring protocol. Gold sentence keys give retrieval recall of 1.000 by construction but do not make the provided response an upper bound on answer quality or faithfulness. Its faithfulness under our claim-support definition is $f = 0.840$ because some benchmark reference responses are not strictly verifiable from

the candidate document set $D_q$. This is consistent with the sparsity of the gold utilised key annotations (Appendix E).

### 5.3   Coverage versus error modes

Retrieval recall measures key-space coverage, while DFNR captures whether the final response still omits required information. MiloNet-full combines high retrieval recall (0.994) with near-zero DFNR, indicating that retrieved evidence is carried into the answer rather than lost during synthesis.

RankGPT and RAPTOR maintain strong key-level coverage but still incur diagnostic false-positive and false-negative flags. DFPR and faithfulness provide complementary views. DFPR records whether any unsupported claim appears, whereas faithfulness measures the supported fraction of claims. MiloNet-full reduces both, showing that coverage alone does not ensure grounded and complete synthesis.

### 5.4   Ablation of MiloNet-core versus MiloNet-full

The core-versus-full ablation evaluates the contribution of MiloNet's complete answer-finalisation stage while holding fixed the upstream evidence pipeline comprising summary-index routing, bounded evidence admission, the SubCoA–folder–document hierarchy, and verified-fact extraction. Both settings perform Stage 2 synthesis over the extracted, document-cited facts, but use different synthesis prompts. MiloNet-core uses the verified-fact synthesis prompt and stops after that call, whereas MiloNet-full uses the constraint-guided synthesis prompt and continues with the final answer checks.

Moving from MiloNet-core to MiloNet-full reduces issue cases from 16 to 2, DFPR from 0.018 to 0.005, and DFNR from 0.038 to 0.003. It also increases faithfulness from 0.850 to 0.938, increases mean answer quality from 2.923 to 2.997, and reduces answer-quality failure rate from 0.038 to 0.003. Because the synthesis prompts differ, this comparison evaluates the complete answer-finalisation stage as a whole rather than isolating the subsequent final answer checks. Under the evaluated RAGBench HotpotQA protocol, MiloNet-core, which does not run the complete answer-finalisation stage, achieves lower DFPR and DFNR than RankGPT and RAPTOR. Its DFPR is 0.018, compared with 0.097 for RankGPT and 0.100 for RAPTOR, and its DFNR is 0.038, compared with 0.095 for RankGPT and 0.087 for RAPTOR. The comparison does not identify which individual upstream component accounts for the core result, but it does show that the complete answer-finalisation stage is not the sole source of MiloNet's diagnostic-error advantage.

Overall, MiloNet-full combines near-zero DFPR and DFNR with the highest faithfulness and mean answer quality under our HotpotQA protocol. The gains over MiloNet-core show the additional value of the complete answer-finalisation stage, while the baseline results show that strong evidence coverage alone does not ensure grounded and complete synthesis.

## 6   Conclusion

This paper presented MiloNet, a framework for traceable RAG in text-grounded settings with a fixed textual evidence corpus. MiloNet combines bounded routing, hierarchical evidence construction, cited synthesis, and final answer checks. On the RAGBench HotpotQA test split under our diagnostic protocol, MiloNet-full achieves near-zero unsupported-claim and omission rates and higher faithfulness than the evaluated baselines. The results support the narrower claim that hierarchical evidence construction, cited synthesis, and final answer checks play complementary roles in reducing unsupported claims and omissions in this benchmark-defined textual evidence setting.

This work has several limitations. Our experiments focus on a single benchmark family, a single benchmark-defined evidence space, and primarily textual evidence, so broader validation across domains, task types, longer contexts, and multimodal settings is still needed. MiloNet also has substantially higher inference-time API cost than simpler baselines. Appendix F reports 127.73 API calls and 474.6k tokens per query for MiloNet-full, compared with 1.00–5.00 calls and 0.4k–3.2k tokens per query for the evaluated baselines. Future work should study cheaper routing, lighter verification policies, sensitivity over the routing caps $K$ and $B$, larger corpus scales, and verification layers adapted to external RAG systems.

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

# Appendix A  Additional notation and implementation details

## A.1  Notation

Let $\mathcal{D} = \{d_i\}_{i=1}^{N}$ denote the document corpus and $\mathcal{T} = \{t_i\}_{i=1}^{N}$ the set of *document-level agent tools* (the leaf tools are LLM-based agents), with a one-to-one mapping $m : \mathcal{T} \to \mathcal{D}$; thus $N = |\mathcal{D}| = |\mathcal{T}|$ in our implementation. Given a query $q$, preprocessing constructs an ordered candidate pool $C(q)$ and a committed tool-filter allowlist $A(q) \subseteq C(q)$ of document-level tool IDs, with a global capacity $|A(q)| \leq K$, where $K$ is the global capacity of the allowlist and $B$ is the per-flush admission cap for tracked tool-call IDs. In our evaluation, we set $K = 30$ and $B = 10$ for all queries. Thus, Algorithm 1 uses $k_1 = \max(1, \lfloor K/3 \rfloor - 2) = 8$ for the initial summary-index pass and $k_2 = K + 2 = 32$ for the enrichment pass. $K$ and $B$ were fixed before evaluation and were not tuned on test outcomes. $K$ bounds the downstream allowlist size, while $B$ bounds the number of planner-tracked IDs admitted at the final tracker flush. We write $u$ for a candidate unit (fact-level intermediate representation) with an associated document identifier $\mathrm{doc}(u)$ when available. Postprocessing admits $u$ for response composition only if $\mathrm{Adm}(u)$ holds; in our implementation, $\mathrm{Adm}(u)$ requires an explicit document identifier $\mathrm{doc}(u)$ and satisfaction of the postprocessing constraints described in §3.3.

## A.2  Preprocessing details

This appendix section provides additional details of the preprocessing stage described in §3.1. MiloNet uses summary-based semantic routing during preprocessing, indexing dense embeddings of document summaries as routing nodes in a shared ChromaDB vector store rather than routing over full document texts. This design reduces topical noise at the routing stage and keeps routing relatively lightweight, as the process defers fine-grained evidence gathering to downstream agents.

An important distinction in our design is that between tool levels, where *document-level tool IDs* refer specifically to tools that invoke document agents (leaf nodes), rather than to higher-level tools (e.g., folder and SubCoA agents) which are not the target of this routing filter. This preprocessing strategy offers two practical advantages. The first is standardisation, in that each source is represented by a fixed-size summary embedding, which reduces sensitivity to document length and helps limit multi-topic dilution in long documents. Second, it improves routing efficiency and stability by filtering irrelevant sources early using summary-level semantics, before performing document-local retrieval and verification.

In our current instantiation, routing nodes correspond to document-level summaries. For substantially longer documents or heterogeneous sources, the same approach can be extended to hierarchical or section-level summaries to preserve routing fidelity without changing the overall three-stage pipeline.

Algorithm 1 summarises how the downstream tool-filter allowlist is constructed from multiple ID sources over the shared summary index. In the algorithm, AppendMeta appends metadata-seeded IDs, extracted from identifier fields in the retrieval metadata of retrieved summary nodes, to the ordered candidate pool before the global truncation step. These identifier fields uniquely identify document-level tools. We refer to

---

**Algorithm 1** Preprocessing for document-level tool-ID allowlist construction.

---

**Require:** user query $q$, summary index $S$, global cap $K$, flush cap $B$
1: $C \leftarrow \emptyset$        ▷ candidate pool $C(q)$
2: $C \leftarrow \mathrm{AppendMeta}(C, \mathrm{IDs}(\mathrm{SummaryQuery}(S, q; k_1)))$    ▷ $k_1 = \max(1, \lfloor K/3 \rfloor - 2)$ (small initial top-$k$)
3: $(y_{\mathrm{coa}}, y_{\mathrm{react}}) \leftarrow (\mathrm{CoA}(q), \mathrm{ReAct}(q))$      ▷ both invoke the tracked summary retriever
4: $\hat{y} \leftarrow \mathrm{SelectAnswerable}(y_{\mathrm{coa}}, y_{\mathrm{react}}, q)$      ▷ CoA→ReAct→ $q$ fallback (text only)
5: $r \leftarrow \mathrm{Canonicalise}(\hat{y})$      ▷ canonicalised search query
6: **if** $r \neq \emptyset$ **then**
7:     $C \leftarrow \mathrm{AppendMeta}(C, \mathrm{IDs}(\mathrm{SummaryQuery}(S, r; k_2)))$      ▷ $k_2 = K+2$ (enrichment top-$k$)
8: **end if**
9: $C \leftarrow \mathrm{Prepend}(\mathrm{Top}_B(\mathrm{TrackedIDs}), C)$    ▷ pooled CoA+ReAct tracked IDs, admitted in one flush
10: $A \leftarrow \mathrm{Truncate}_K(\mathrm{OrderDedupe}(C))$      ▷ committed allowlist $A(q)$
11: $q' \leftarrow \mathrm{RefineQuery}(q, r)$      ▷ optional "Additional focus"
12: **return** allowlist $A$ and refined query $q'$

---

this intermediate, growing ordered pool of candidates as the *candidate ID pool $C(q)$*. After candidates from multiple ID sources are concatenated, de-duplicated in order-preserving fashion, and truncated to the global cap ($|A(q)| \leq K$), the committed list $A(q)$ is used as the *tool-filter allowlist* for downstream agents.

The procedure begins with a summary-index query over the original user input, which seeds candidate *document-level tool IDs*. Before further filtering occurs, the candidate ID pool contains only these initial summary-seeded metadata IDs (IDs recorded by tracked tool calls remain buffered and are not flushed into the pool at this stage). This first seeding pass uses a small top-$k$, set to approximately one third of the final allowlist capacity plus a small offset, for example, $-2$, subject to a minimum of one, to keep initial routing cheap and stable. The same user input query is also forwarded to the CoA-style and ReAct-style planners.

Both planners produce textual responses and are equipped with the same tracking retriever built over the summary index. These planner-internal retriever calls are distinct from the metadata-only summary-index passes in Fig. 1. Localised context snippets (a two-sentence context window, truncated to at most 1200 characters) are produced only when the CoA and ReAct planners invoke this tracking retriever during planning. In that pathway, retrieved nodes are post-processed into snippets to provide lightweight context while the tracker records candidate document-level tool IDs for subsequent flushing.

MiloNet adopts a fallback policy that leverages two complementary planners, since CoA-style and ReAct-style planning can fail in different ways, especially when tool interaction is required. For each query, MiloNet runs both planners to produce two textual draft responses, and then uses a lightweight LLM-based selector to assess whether either draft answers the query. If the CoA draft is judged answerable, MiloNet keeps it. Otherwise, if the ReAct draft is judged answerable, MiloNet falls back to the ReAct draft. If neither is judged answerable, MiloNet reverts to the original user query.

The selected draft is then rewritten by the same lightweight LLM into a search query, canonicalised as `search_query`. In this context, canonicalisation preserves explicit informational keywords and removes tool IDs, variables, and other non-informational tokens from the selected draft. MiloNet uses `search_query` for auxiliary summary lookup whenever a draft response is available, including cases where the fallback policy reverts to the original user query.

Importantly, the selector gates only the initial *textual* response used for canonicalisation and downstream lookup. Tracked tool-call IDs from both planners are pooled in a pending buffer and added to the candidate ID pool later during the tracker flush, before order-preserving de-duplication and truncation to the global cap. Empirically, the two planners show complementary failure modes. The CoA-style planner can be more stable on queries where iterative tool use may cycle. The ReAct-style planner can recover relevant sources when the CoA planner misses them because of suboptimal keyword selection or query reformulation.

Using this `search_query`, the system may perform an **auxiliary summary-index retrieval pass** to enrich the *candidate ID pool* before downstream execution. This auxiliary retrieval is the *second* summary-index pass in Fig. 1, which runs after canonicalisation with an intentionally larger top-$k$, set slightly above the

final allowlist capacity, for example, capacity $+2$, to improve recall during enrichment. Query refinement is then applied *after* the second auxiliary summary lookup. A lightweight LLM prompted for query refinement assesses whether the information in `search_query` is sufficiently complete and unambiguous. If it is incomplete or uncertain, the system retains the original user query. Otherwise, it outputs a two-line refined query consisting of the original query and an `Additional focus: ...` line that reuses only salient keywords from `search_query` such as names, dates, locations, and numeric values. When the refined query differs non-trivially, it replaces the user input for the subsequent core RAG execution. Otherwise, the original query is retained.

The routing stage maintains a growing ordered *candidate ID pool* and a committed **tool-filter allowlist**. Tool-call tracking accumulates pending IDs during planner execution. At a single flush at the end of preprocessing, the tracker admits up to 10 pending IDs using an internal priority that combines (i) the best similarity score observed for the ID, (ii) hit frequency, (iii) keyword-hit signals, and (iv) a recency-based penalty. These top-10 IDs are prepended to the candidate ID pool. Metadata-seeded IDs from the two summary-index passes are appended without a comparable score. The pool is then de-duplicated in order-preserving fashion and truncated to a **configurable global capacity** to form the final allowlist for downstream agents. This gives two-tier capacity control, as shown in Fig. 1. First, tracked tool calls are subject to a per-flush admission cap. Second, global truncation after concatenation and de-duplication enforces $|A(q)| \leq K$.

To preserve candidate diversity and recall, we intentionally avoid a single global re-ranking over all candidates. First, candidates come from heterogeneous sources with signals that are not directly comparable, for example, metadata-seeded IDs do not carry a meaningful score. Second, summary-level similarity is only an imperfect proxy for downstream usefulness. For entity- or keyword-critical queries, some important sources may receive low summary similarity even when they contain decisive evidence.

Preprocessing emits (i) a tool-filter allowlist and (ii) an optional refined user input. This stage is therefore **non-assertive**. It does not produce user-facing factual claims that enter the final response. Instead, it produces only routing constraints, namely tool eligibility through the tool-filter allowlist, which determine *where* evidence will be retrieved in later stages before any evidence-grounded composition takes place.

### A.3 Core implementation details

This appendix section provides additional details of the core RAG stage described in §3.2. As shown in Fig. 2, MiloNet performs core retrieval and evidence construction through a hierarchical agent structure. The top layer consists of multiple **SubCoA agents**, each coordinating a fixed set of **folder agents**. In our implementation, each SubCoA agent contains **64** folder agents, and each folder agent manages at most **3 document agents**.[3] In our experiments, the $\leq 3$ documents-per-folder property is provided by the processed corpus snapshot (constructed upstream) and is not enforced by MiloNet during initialisation. SubCoA and folder agents act as CoA-style controllers for planning and aggregation, whereas document agents are leaf-level tools that execute document-local retrieval and candidate extraction.

**Runtime agent registry** At system startup, MiloNet loads the preconstructed processed-corpus snapshot and instantiates the folder and document agents recorded in its fixed registry, exposing them as callable tools to the runtime system. This component performs one-time loading and state management, while all query-time behaviour follows the same preprocessing-core-postprocessing pipeline described above. To satisfy tool-interface constraints (e.g., limits on tool name length), MiloNet assigns compact, stable identifiers to folder agents and document-level tools, and applies collision handling (via an incremental suffix when needed) to ensure uniqueness of document-level tool IDs. These identifiers are used consistently during tool registration and routing.

The runtime registry maintains the parent-child association between folder agents and document-level tools. At query time, SubCoA applies the preprocessing-derived allowlist by mapping allowed document-level tool IDs to their parent folder agents, activating only the corresponding folder agents, and passing each folder

---

[3]This cap is imposed by tool-interface length constraints. Registering too many document tools within a single agent can exceed tool description length limits and fail, preventing downstream routing to the corresponding document agents.

agent the restricted tool subset. Each folder agent then orchestrates retrieval within this subset, aggregates candidate units, and forwards a structured intermediate state back to SubCoA for higher-level composition.

**Reference disambiguation for verified facts**  A small LLM rewrites fact sentences using the originating document context to replace ambiguous pronouns (e.g., *it*, *they*, and possessive forms) with explicit entity mentions when available. This rewriting is *faithfulness-preserving* in that it performs only subject and reference resolution, does not introduce new information, and returns the original sentence unchanged when no ambiguity is detected. The resulting disambiguated verified facts reduce cross-sentence ambiguity during aggregation and mitigate spurious associations when composing multi-source answers.

**Parallel evidence construction**  During parallel execution, the same wrapped query string, derived from the original or enhanced query, is passed through the hierarchy without per-agent rewriting. The dispatch is therefore *input-invariant*. Differences in retrieved evidence come from agent-specific context, including folder summaries, document metadata, and document-local indices.

In implementation, each document agent includes a local vector index for intra-document retrieval together with a node-level postprocessing pipeline that converts retrieved snippets into structured **candidate units**. This pipeline includes the default LlamaIndex `SimilarityPostprocessor` to filter low-relevance retrieved nodes and a logging-only trace for observability and analysis during experiments, which does not affect retrieval outputs or verification results.

Candidate units are intermediate representations, for example, extracted atomic fact statements paired with provenance information, including the originating document ID and the evidence snippet(s) supporting the statement. They are treated as *proposals* rather than trusted facts, are not directly emitted to the user, and must pass downstream postprocessing and verification before they are eligible for response composition.

These components of the core structured RAG stage turn preprocessing-level routing artifacts (a constrained allowlist of eligible document-level tool IDs) into a set of candidate units cited by document IDs, suitable for verification-driven postprocessing.

## A.4  Postprocessing details

This appendix section provides additional details of the postprocessing stage described in §3.3. MiloNet treats outputs from the core structured RAG stage as candidate units and applies a sequence of downstream checks to determine which units are admissible for response composition. We write $\mathrm{Adm}(u)$ for the admissibility predicate induced by postprocessing. In our implementation, $\mathrm{Adm}(u)$ requires an explicit document identifier $\mathrm{doc}(u)$ together with the downstream constraint checks described below.

**Stage 1 (candidate consolidation and citation preparation)**  At this stage, candidate units from multiple SubCoA agents are normalised and associated with explicit evidence pointers, namely document IDs used as citations. Candidate units may also be lightly rewritten to reduce ambiguity, for example, by resolving unclear references, while preserving faithfulness to the content provided by the SubCoA agents.

**Stage 2 (cited answer synthesis)**  Both MiloNet-core and MiloNet-full perform Stage 2 synthesis over the extracted, document-cited facts. The verified-fact synthesis prompt used by MiloNet-core asks the model to compose a direct answer from these facts, after which MiloNet-core stops. The constraint-guided synthesis prompt used by MiloNet-full adds explicit evidence, citation, scope, formatting, and anti-hallucination instructions before proceeding to the final answer-checking layer.

**Admissibility and neutral handling of missing evidence**  After Stage 2 synthesis, MiloNet-full checks whether the composed response satisfies the required evidence, citation, scope, formatting, and anti-hallucination constraints. In particular, if a supporting-details bullet is emitted without an explicit document-ID citation, which is a rare rule violation, it is detected and explicitly flagged as uncited in subsequent postprocessing passes.

MiloNet treats a composite conclusion as a collection of intermediate claims and requires each claim to be individually supported by evidence-linked candidate units (with citations). If any required step lacks sufficient support, the system does not emit the composite conclusion. When evidence is insufficient, MiloNet treats missing support as neutral and returns either a partial answer consisting only of document-cited statements or explicitly states that the provided documents do not contain sufficient evidence to answer the question, rather than filling gaps with unsupported inference or hallucination.

As shown in Fig. 3, the detailed postprocessing pipeline (i) extracts and composes answers from candidate units, (ii) enforces citation completeness and flags uncited statements as unverified (i.e., bullets in supporting details that do not match the document-ID citation pattern), and (iii) applies scope and attribute checks and fluency-only rewriting without adding new information.

Concretely, *supporting-details citation completeness* scans the bullets in the supporting details and requires each top-level bullet block (including its nested lines) to contain at least one document ID; otherwise it appends a fixed disclaimer stating that the statement is not cited in the provided documents. *Uncited claim flagging* propagates this signal to the opening line of the response. If any bullet in the supporting details carries the uncited disclaimer, this stage appends a fixed annotation indicating that the overall conclusion relies on information not cited in the provided documents. *Verified fact reinforcement* checks whether folder-level verified facts are present in the SubCoA input context. It normalises each fact by stripping any trailing parenthesised evidence quote and performs a case-insensitive substring containment check against the current response. If any normalised facts are absent, it appends the corresponding original fact lines verbatim to the end of the response, restoring document-cited positive evidence that was dropped during synthesis or formatting.

*Scope guard* validates the opening answer against the supporting details using a lightweight LLM validator under strict scope, timeframe, and evidence constraints. When misalignment is detected, it rewrites only the opening answer to align with the cited evidence. Supporting details are treated as verbatim evidence and are not rewritten. The validation loop runs for up to two rewrite and revalidation attempts to enforce coverage of the query's requested attribute keyword. If the validator indicates missing support by returning document IDs, the stage augments the supporting details by injecting the corresponding supporting sentences before re-validating. If the validator fails to return a valid structured output, the stage falls back to leaving the response unchanged or applying a minimal rewrite to the opening answer only. Finally, deterministic textual cleanup is applied (e.g., removing residual interrogative phrasing and normalising quotation marks) without introducing new factual content; if cleanup yields an empty string, the stage falls back to the uncleaned validator output.

*Citation formatting* normalises citation format by wrapping bare document IDs into parenthesised citations, e.g., abcd_1; abcd_2 → (abcd_1; abcd_2), leaving already-parenthesised identifiers unchanged. *Citation deduplication* then operates on these parenthesised citation groups, collapsing duplicate document identifiers within each group while preserving first-occurrence order and normalising separators (e.g., to semicolons). *Attribute completeness* is a deterministic fallback that runs only when the preceding scope validator has not applied a rewrite. If the lead answer portion of the response omits the requested attribute keyword, it selects a supporting-details sentence (preferring one that contains the keyword), cleans it, and overwrites the lead answer with that sentence, applying only quotation and punctuation fixes. This step does not append or rewrite the supporting details and does not re-check scope or timeframe. It only ensures that the requested attribute is stated explicitly in the answer.

*Fluency smoothing* is the final non-assertive polishing stage and runs only when the previous scope validator has not rewritten the response. It rewrites the lead answer portion of the response for clarity using a lightweight LLM. The rewrite is guided by the supporting details, and to prevent content drift, the rewritten answer is accepted only if it maintains sufficient lexical overlap with the original answer and the supporting details; otherwise the original answer is retained. Finally, it removes extra whitespace and balances quotation marks. During this stage, the supporting details in the response are treated as verbatim evidence and not rewritten.

After these postprocessing stages, the final response is assembled from admissible candidate units with document-ID citations, while any uncited statements are explicitly flagged.

## A.5  Processed-corpus construction and evaluation materials

Figure 4 summarises the processed-corpus construction pipeline used before query-time inference.

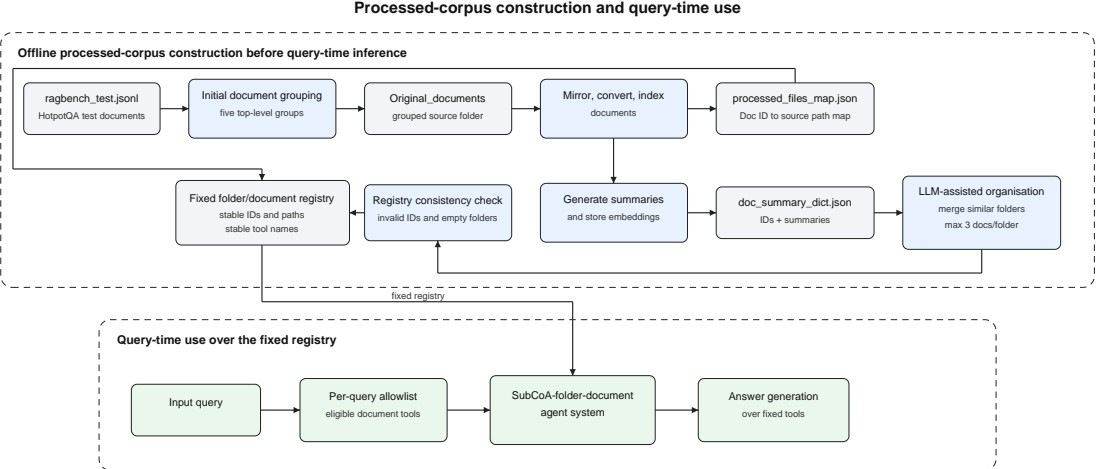

Figure 4: Processed-corpus construction and query-time use. The folder and document registry is constructed once before evaluation and remains fixed during query-time inference.

The processed corpus is constructed once before query-time inference. Starting from `ragbench_test.jsonl`, we extract the source documents for the HotpotQA test split and organise them into five top-level groups for this processed snapshot. The resulting original-document folder is then converted into the folder and document registry used by MiloNet. During initialisation, MiloNet builds a mirror folder structure, indexes each document, generates a document summary, stores document-summary embeddings in ChromaDB, and writes `processed_files_map.json`. This map links compact document IDs back to their source document paths, allowing the final fixed folder and document registry to be materialised with stable IDs, paths, and tool names. From the Chroma metadata, we extract document IDs and document summaries into `doc_summary_dict.json`. An LLM-assisted organisation step groups semantically similar documents into folders to produce the fixed processed snapshot used by MiloNet. A final consistency check compares document IDs referenced by the organised folder structure against the document-summary records, identifies missing or extra IDs, and removes invalid file nodes or empty folders before the final folder and document registry is materialised.

This processed corpus snapshot is fixed before inference and is used unchanged for all MiloNet queries. Query-time routing and answer generation do not modify the folder and document hierarchy. The hierarchy therefore defines the tool registry available to the SubCoA-folder-document agent system, while the per-query allowlist described in §3.1 controls which document-level tools are eligible for a given query. The folder and document registry acts as a corpus-level interface. Applying MiloNet to a new fixed textual corpus requires constructing the corresponding processed registry before query-time inference.

The main effectiveness results use the 390-query HotpotQA test split from RAGBench and the pooled corpus described in §4.1. For each Table 3 row, we record the system output used for that row, adapter mapping, diagnostic labels, answer-quality rubric scores, and faithfulness scores.

## Appendix B    RAPTOR configuration selection

RAPTOR introduces two configuration parameters controlling the number of clusters selected and the number of retained nodes per query. We evaluated a grid of configurations. Here, $c$ denotes the number of selected clusters and $k$ denotes the global top-$k$ node budget used for context assembly. We evaluated `c2_k4`, `c2_k6`,

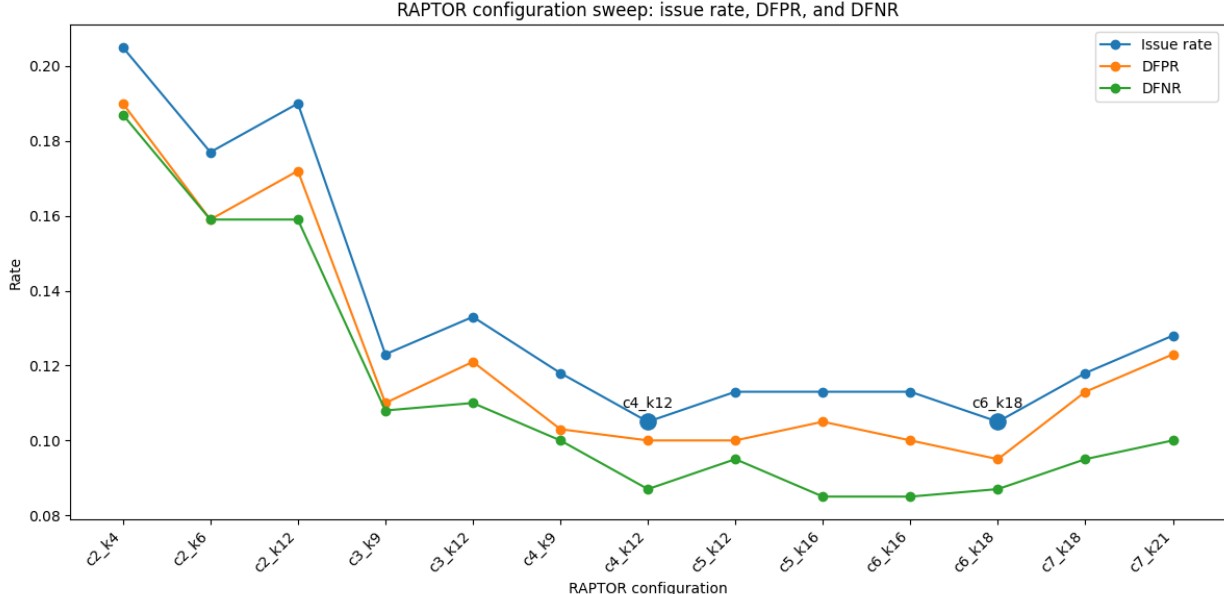

Figure 5: RAPTOR configuration sweep on the HotpotQA test split (390 queries). We report the issue rate, diagnostic false-positive rate (DFPR), and diagnostic false-negative rate (DFNR) for each $(c, k)$ configuration, where $c$ is the number of selected clusters and $k$ is the global top-$k$ node budget used for context assembly. Configurations `c4_k12` and `c6_k18` are highlighted.

`c2_k12`, `c3_k9`, `c3_k12`, `c4_k9`, `c4_k12`, `c5_k12`, `c5_k16`, `c6_k16`, `c6_k18`, `c7_k18`, and `c7_k21`, and applied the same evaluation pipeline and diagnostics to each run.

We define an *issue case* as a query for which the diagnostic process flags at least one diagnostic false positive or at least one diagnostic false negative. Figure 5 plots the resulting issue rate, DFPR, and DFNR across configurations. Figure 5 shows a steep reduction in these metrics when the configuration moves from the c2_* regime to c4_*, after which the curves largely flatten. We select `c4_k12` as a mid-range configuration with low issue rate (41 out of 390), DFPR (39 out of 390), and DFNR (34 out of 390). Larger configurations provide limited gains. `c6_k18` matches the same issue count while requiring larger $c$ and $k$, and some larger settings increase error rates. We therefore report RAPTOR results using `c4_k12`.

## Appendix C  Alignment and supplementary metric definitions

All alignment metrics are computed in the sentence-key space after adapter mapping. For each query $q$, we use the same notation in §4.4: $G_q$ (gold relevant sentence keys), $R_q$ (retrieved sentence keys), $U_q$ (gold utilised sentence keys), and $S_q$ (sentence keys used by the system). For the reference baseline, we set $R_q := G_q$ and $S_q := U_q$ by construction.

We define retrieval precision and retrieval recall as

$$rp = \frac{|R_q \cap G_q|}{|R_q|}, \qquad rr = \frac{|R_q \cap G_q|}{|G_q|},$$

and define retrieval F1 as

$$r\_f1 = \frac{2\, rp\, rr}{rp + rr + \varepsilon},$$

where $\varepsilon$ is a small constant.

| System | $rp$ | $r\_f1$ | $gp$ | $g\_f1$ | trustworthiness | correctness |
|---|---|---|---|---|---|---|
| MiloNet-full | 0.306 | 0.442 | 0.409 | 0.539 | 0.965 | 0.639 |
| MiloNet-core | 0.290 | 0.425 | 0.435 | 0.553 | 0.924 | 0.609 |
| Vanilla RAG | 0.338 | 0.424 | 0.254 | 0.348 | 0.688 | 0.507 |
| RankGPT | 0.530 | 0.633 | 0.396 | 0.519 | 0.894 | 0.670 |
| RAPTOR c4\_k12 | 0.221 | 0.352 | 0.156 | 0.262 | 0.945 | 0.490 |
| RAGBench reference baseline | 1.000 | 1.000 | $0.982^{\dagger}$ | $0.982^{\dagger}$ | 0.941 | 0.941 |

Table 4: Supplementary alignment and composite metrics on the HotpotQA test split (390 queries). Precision-based terms ($rp$, $gp$, $r\_f1$, $g\_f1$, and correctness) are sensitive to key extraction and benchmark constraints, and should be interpreted together with Appendix D and Appendix E. Trustworthiness is reported here as a supplementary composite diagnostic rather than as a main-table metric. $^{\dagger}$RAGBench contains 7 instances with an empty gold utilised set ($|U_q| = 0$). Under our convention, empty-denominator ratios are defined as zero, yielding macro-averaged generation-side scores below 1.

We define generation precision and generation recall as

$$gp = \frac{|S_q \cap U_q|}{|S_q|}, \qquad gr = \frac{|S_q \cap U_q|}{|U_q|},$$

and define generation F1 as

$$g\_f1 = \frac{2\, gp\, gr}{gp + gr + \varepsilon}.$$

When a denominator is empty, the corresponding ratio is defined as zero.

Based on these components and the faithfulness score $f$ defined in §4.4, we report two composite diagnostics:

$$\text{trustworthiness} = \frac{rr + gr + f}{3}, \qquad \text{correctness} = \frac{r\_f1 + g\_f1 + f}{3}.$$

These composite scores are supplementary diagnostics and are not used as primary evidence in the main comparison.

Because precision depends on the extracted denominators $|R_q|$ and $|S_q|$, precision-based metrics ($rp$, $gp$, $r\_f1$, $g\_f1$, and correctness) are sensitive to key-extraction and to evaluation constraints such as candidate-set coverage ($D_q$) and granularity mapping (document-to-sentence expansion), and should be interpreted together with Appendix D and Appendix E. As a supplement, Table 4 reports the precision-based and composite diagnostics for all systems.

## Appendix D  Retrieval precision interpretation

Retrieval precision ($rp$) is sensitive to both benchmark scope and how systems are mapped to sentence keys. First, MiloNet performs corpus-scale routing over a pooled corpus ($N$=1558), while most baselines operate on the per-query candidate set $D_q$. Second, for MiloNet, retrieved sentence keys are obtained by matching benchmark sentences appearing in the retrieved document context blocks. However, RankGPT and RAPTOR select at document granularity, and the selected documents are deterministically expanded to sentence keys for evaluation, which changes the effective denominator in precision. Third, gold annotations cover only $D_q$, so corpus-scale retrievals that are semantically correct but outside $D_q$ are penalised under this benchmark setting. Accordingly, $rr$ and $rp$ should be read as coverage and efficiency with respect to the benchmark candidate scope (i.e., $G_q$ defined over $D_q$), not the full pooled corpus. Fourth, high retrieval precision does not guarantee adequacy. A system can retrieve a small set of sentence keys that are mostly gold relevant (high $rp$) while omitting required relevant sentence keys, which is better reflected by DFNR and $rr$. We report retrieval precision for completeness but emphasise recall-based coverage ($rr$) and DFNR for retrieval adequacy, and faithfulness for end-to-end grounding.

## Appendix E  Generation precision interpretation

Generation precision ($gp$) is primarily an efficiency measure under sparse gold utilisation annotations and is sensitive to key extraction. First, gold utilised sentence keys are typically sparse under the benchmark annotation. Thus, $gp$ can be low even for well-grounded responses that draw on additional document-supported sentences beyond the gold utilised set. Second, generation precision is sensitive to how the used sentence-key set $S_q$ is extracted. In our pipeline, $S_q$ is derived from sentence keys matched to the content provided to the generator, which can include sentences present in retrieved contexts that are not ultimately expressed as atomic claims in the final response. Third, extraction procedures can differ across systems due to differences in output format and context construction, which affects the effective denominator in $gp$. Moreover, high generation precision does not preclude omissions or unsupported claims. $gp$ is a precision term over the sparse gold utilised set and does not capture coverage ($gr$) or claim-level support. Accordingly, a system can achieve high $gp$ while still omitting utilised keys (low $gr$) or producing unsupported claims (low $f$ and high DFPR). For these reasons, we do not treat generation precision as a primary indicator of reliability. The main analysis instead emphasises faithfulness, DFPR, DFNR, mean answer quality, and answer-quality failure rate.

## Appendix F  Inference-time cost profile

We profile inference-time API cost on the first 100 RAGBench HotpotQA test cases. The profile excludes training, corpus preprocessing, and one-time indexing. We report API calls per query and total tokens per query as the primary inference-cost indicators. Raw observed API latency is not treated as controlled runtime because the implementation uses asynchronous API-backed agent execution and provider-side latency varies across calls.

| System | $n$ | API calls per query ↓ | Total tokens per query ↓ |
|---|---|---|---|
| Vanilla RAG | 100 | 1.00 | 410.9 |
| RankGPT | 100 | 2.00 | 856.4 |
| RAPTOR c4_k12 | 100 | 5.00 | 3,226.8 |
| MiloNet-core | 100 | 127.04 | 455,798.5 |
| MiloNet-full | 100 | 127.73 | 474,590.0 |
| Full–core delta | 100 | +0.69 | +18,791.5 |

Table 5: Inference-time API cost profile on the first 100 RAGBench HotpotQA test cases. The profile excludes training, corpus preprocessing, and one-time indexing.

MiloNet is substantially more expensive than the evaluated baselines. Moving from MiloNet-core to MiloNet-full adds 0.69 API calls and about 18.8k tokens per query. This net core-to-full difference is associated with the complete answer-finalisation stage. Most of MiloNet's total inference cost is associated with the shared routing, hierarchical evidence construction, and verified-fact extraction stages, while the complete answer-finalisation stage adds a smaller marginal overhead relative to the shared pipeline.

