# OpenReview forum: "MiloNet: A Framework for Traceable and Verified RAG"
_TMLR — Under review for TMLR_

### Review · Reviewer_vmE2 · 2026-06-12

**Summary Of Contributions:**

The paper proposes MiloNet, a retrieval-augmented generation framework focused on improving reliability through explicit control of evidence flow and answer construction. The framework combines three main components: (i) a summary-based semantic routing that develops a strict list of documents that can be retrieved usina a COA and a ReAct planner, (ii) a hierarchical retrieval architecture consisting of SubCoA, folder and document agents that construct document-cited candidate facts,  and (iii) a two stage postprocessing framework that filters unsupported content and constrains the final answer. The paper also introduces a unified evaluation protocol on the RAGBench HotpotQA benchmark and reports substantial reductions in unsupported claims and omissions relative to several RAG baselines.

**Audience:**

Yes

**Audience Explanation:**

Reliability, hallucination reduction, attribution, and verification remain central challenges in retrieval-augmented generation. The paper proposes a practical systems-oriented approach that focuses on controlling evidence admission and answer synthesis rather than solely improving retrieval quality. The distinction between evidence coverage and grounded synthesis is interesting, and the framework provides a concrete example of how verification and provenance constraints can be integrated into modern RAG pipelines. Researchers and practitioners working on RAG systems, trustworthy AI, LLM evaluation, and production deployment would likely find the findings relevant.

**Claims And Evidence:**

No

**Claims Explanation:**

The experimental results demonstrate that the proposed pipeline achieves strong performance on the selected benchmark and that the postprocessing stage contributes significantly to reducing hallucinations and omissions. The comparison between MiloNet-core and MiloNet-full provides useful evidence regarding the contribution of the verification layer.

However, several aspects limit the strength of the conclusions:

- Evaluation is restricted to a single benchmark family (HotpotQA within RAGBench). The central claim is that the framework enables "decision-grade" reliability, but the experiments are conducted only on one dataset and one task type. It is unclear whether the observed improvements would generalize to other QA benchmarks, open-domain retrieval settings, enterprise document collections, or long-context tasks.

- The proposed method combines many interacting components which renders it hard to follow. While the paper includes an ablation comparing MiloNet-core and MiloNet-full, it does not isolate the individual contributions of the routing stage, hierarchical agent structure, planner selection, query refinement, allowlist construction, or specific postprocessing modules. Consequently, it remains difficult to determine which components are responsible for the reported gains.

- The computational trade-offs are insufficiently analyzed. The framework introduces multiple retrieval stages, hierarchical agents, verification passes, and postprocessing modules. A detailed analysis on inference cost and scalability would strengthen the practical significance of the results. The paper acknowledges increased inference cost but does not provide a systematic evaluation of this trade-off.

**Requested Changes:**

- Expand the empirical evaluation beyond a single benchmark. Include additional datasets and task types (e.g., other QA benchmarks, long-document retrieval tasks, enterprise-style corpora, or multi-document reasoning benchmarks) to assess whether the reliability gains generalize beyond HotpotQA.

- Provide finer-grained ablation studies. Isolate the contributions of the major components of MiloNet, including summary-based routing, hierarchical retrieval, planner selection, query refinement, allowlist construction, and the individual verification/postprocessing modules. This would clarify which elements are most responsible for the improvements.

- Analyze computational cost and scalability. Report latency, token consumption, number of model calls, and runtime breakdowns for the different stages of the pipeline. Since MiloNet introduces multiple retrieval and verification layers, understanding the reliability-versus-cost trade-off is important for practical deployment.

---

> ### Author Response · Authors · 2026-07-11
> **Response to Reviewer vmE2 (1/3): Evaluation scope beyond a single benchmark**
>
> > Reviewer comment: "Evaluation is restricted to a single benchmark family (HotpotQA within RAGBench). The central claim is that the framework enables 'decision-grade' reliability, but the experiments are conducted only on one dataset and one task type. It is unclear whether the observed improvements would generalize to other QA benchmarks, open-domain retrieval settings, enterprise document collections, or long-context tasks."
> >
> > Requested change: "Expand the empirical evaluation beyond a single benchmark. Include additional datasets and task types (e.g., other QA benchmarks, long-document retrieval tasks, enterprise-style corpora, or multi-document reasoning benchmarks) to assess whether the reliability gains generalize beyond HotpotQA."
>
> We agree that establishing generalisation beyond HotpotQA requires evaluation on additional benchmarks. As this revision does not report a second benchmark, we limit the empirical claim to the evaluated RAGBench HotpotQA setting. MiloNet's hierarchy is designed as a corpus-level interface rather than a HotpotQA-specific rule, although each new benchmark or deployment setting requires a corresponding processed registry, document summaries, tool interfaces, and adapter mappings for the new evidence space. The Introduction now frames MiloNet as a framework for text-grounded RAG over a fixed textual evidence corpus, and Appendix A.5 documents how the registry used in the reported evaluation is constructed.
>
> Within this scope, the current evidence supports a clear empirical claim. On the 390-query RAGBench HotpotQA split, the reported results show lower diagnostic error rates, higher answer quality, and higher faithfulness for MiloNet-full under the shared downstream evaluation protocol. Deterministic system-specific adapters are used for sentence-key alignment. These results are presented as evidence for the evaluated benchmark-defined text-grounded setting.
>
> To reflect this scope, we changed the earlier Abstract claim that "decision-grade RAG requires explicit control" over synthesis, provenance, and admissibility. The revised Abstract states that, in the evaluated RAGBench HotpotQA setting, hierarchical evidence construction, cited synthesis, and final answer checks help reduce unsupported claims and omissions. Section 6 uses the same framing and states the limits of the current empirical evidence.
>
> Original location: the Abstract, Introduction, and conclusion made broader reliability claims (original pp.1-2 and p.12). Revised location: the Abstract, Section 1, and Section 6 now limit the empirical claim to the evaluated RAGBench HotpotQA setting (revised pp.1-2 and p.12).

---

> > ### Author Response · Authors · 2026-07-11
> > **Response to Reviewer vmE2 (2/3): Component attribution and ablations**
> >
> > > Reviewer comment: "The proposed method combines many interacting components which renders it hard to follow. While the paper includes an ablation comparing MiloNet-core and MiloNet-full, it does not isolate the individual contributions of the routing stage, hierarchical agent structure, planner selection, query refinement, allowlist construction, or specific postprocessing modules. Consequently, it remains difficult to determine which components are responsible for the reported gains."
> > >
> > > Requested change: "Provide finer-grained ablation studies. Isolate the contributions of the major components of MiloNet, including summary-based routing, hierarchical retrieval, planner selection, query refinement, allowlist construction, and the individual verification/postprocessing modules. This would clarify which elements are most responsible for the improvements."
> >
> > We agree that attributing the reported gains to individual components requires finer-grained ablations. The present revision does not add separate ablations for each upstream and checking component. The existing comparison answers a narrower question by evaluating the complete answer-finalisation stage as a whole. Both variants share the same upstream evidence pipeline, including summary-index routing, bounded evidence admission, the SubCoA-folder-document hierarchy, and verified-fact extraction. Both perform Stage 2 synthesis over extracted, document-cited facts, but use different synthesis prompts. MiloNet-core uses the verified-fact synthesis prompt and stops after that call, whereas MiloNet-full uses the constraint-guided synthesis prompt and then applies the subsequent final answer checks. Because prompt selection is part of the complete answer-finalisation stage, the core and full comparison evaluates this stage as a whole rather than the contribution of each individual check.
> >
> > Under the evaluated RAGBench HotpotQA protocol, MiloNet-core, which does not run the complete answer-finalisation stage, achieves lower DFPR and DFNR than RankGPT and RAPTOR. MiloNet-full has lower diagnostic error rates, with DFPR=0.005 and DFNR=0.003. The comparison does not identify which individual upstream component accounts for the core result, but it does show that the complete answer-finalisation stage is not the sole source of MiloNet's diagnostic-error advantage.
> >
> > Original location: Section 3.3, the original Figure 3 caption, Section 4.2, and Section 5.4 did not draw the ablation boundary tightly enough (original pp.5-8 and p.12). Revised location: Section 3.3 describes the synthesis and checking pipeline, while the revised Figure 3 caption, Table 1, Section 4.2, and Section 5.4 present the core and full comparison as a stage-level ablation of the complete answer-finalisation stage and avoid assigning the result to individual upstream or checking components (revised pp.5-8 and p.12). Appendix A.4 describes the two synthesis prompts and subsequent final answer checks (revised pp.17-18). Section 6 separately identifies verification layers adapted to external RAG systems as future work (revised p.12).

---

> > > ### Author Response · Authors · 2026-07-11
> > > **Response to Reviewer vmE2 (3/3): Computational cost and scalability**
> > >
> > > > Reviewer comment: "The computational trade-offs are insufficiently analyzed. The framework introduces multiple retrieval stages, hierarchical agents, verification passes, and postprocessing modules. A detailed analysis on inference cost and scalability would strengthen the practical significance of the results."
> > > >
> > > > Requested change: "Analyze computational cost and scalability. Report latency, token consumption, number of model calls, and runtime breakdowns for the different stages of the pipeline. Since MiloNet introduces multiple retrieval and verification layers, understanding the reliability-versus-cost trade-off is important for practical deployment."
> > >
> > > In response, we added a 100-query inference-time cost profile in Appendix F. It reports API calls per query and total tokens per query, shows that MiloNet has substantially higher absolute API cost than the simpler baselines, and records an increase of 0.69 calls and approximately 18.8k tokens per query from the core setting to the full setting. Because MiloNet uses asynchronous API-backed execution and provider-side latency varies across calls, we use calls and token usage as reproducible cost indicators rather than interpreting raw latency as controlled runtime. This net core-to-full difference is associated with the complete answer-finalisation stage. Stage-level wall-clock attribution would require timing under a stable execution environment, so the present analysis focuses on the API-use quantities that can be compared consistently across systems.
> > >
> > > We agree that the evaluated 1,558-document corpus does not establish industrial-scale scalability. The revised manuscript therefore limits the scalability discussion to the demonstrated use of bounded routing to control the candidate evidence space over this pooled corpus.
> > >
> > > In the original manuscript, the method overview described MiloNet as scalable and referred to routing over a large corpus (original pp.2-3). Inference cost was mentioned only qualitatively as a limitation, without a quantitative cost profile (original p.12). In the revised manuscript, Section 3 narrows the bounded-routing claim, Appendix F reports the inference-time cost profile, and Section 6 summarises the resulting cost and scale limitations (revised pp.2-3, p.12, and p.22).

---

### Review · Reviewer_4aVE · 2026-06-13

**Summary Of Contributions:**

This paper present a three stage inference RAG pipeline for LLM to achieve "Decision grade" RAG. Firstly, his paper propose to embeds doc summaries in ChromaDB and builds a bounded allowlist with semantic routing. Secondly, the paper introduce a hierarchical SubCoA with folder/document agent. Each folder agent sends queries to document agents with restricted tool callings. Thirdly, the paper introduce a sequence of constraint checks as a post-processing for the answers. This ensures answers with explicit document can pass the check. The submission also builds a unified evaluation adapter to evaluate on RAG bench. The papers shows the proposed system outperform other benchmarks and the oracle setting on most metrics when evaluated with their adapters.

To sum up the submission have the following strengths:
- This paper presents a structured way to strengthen LLMs to generate more grounded and verifiable output, taming the typical behavior of hallucination and presenting unfounded citations.
- The proposed method is verified by their modified RAGBench HotpotQA test split. The model achieved better performance then previous papers on the proposed setup.

Limitations:
- This is only a post-processing pipeline on a trained model. The performance of the underlying model and their "cooperativeness" of the trained model might matter more than the framework itself.
- The quantitive evaluation of the proposed method is not comprehensive and fully convincing. The authors evaluated on a modified setup of RAGBench with their proposed adapters for this task. Different models are using different backbone models. The proposed method achieved better performance than oracle model. On the way hand it suggests its a strong performance but one may also question if the evaluation metric is distinctive enough.
- The proposed model's post-processing is explicitly guaranteed to perform better on the proposed metrics as the metrics mostly check if the citations are founded and from an actual document. However, the reviewer did not see direct metrics on the actual quality on the answers. Presumably the answers with valid citations should be better but they are not always a consistently equivalent evaluation metrics.
- The proposed methods have some hyper-parameters such as global cap K, flush cap B but there is no ablative study on its sensitivity and impacts on the performance.
- The system is not easy to reproduce as both the evaluation metric and the methods are not open sourced.
- The authors mentioned the proposed method would have higher inference cost but did not quantitatively study it and it's unclear about the compute vs performance trade off of the proposed systems.
- The evaluation metric like "trustworthiness" and "correctness" seem a bit arbitrary. Do we have sufficient study to show the metric are sound? The evaluation metric itself sound arbitrary without rigorous study or reasoning.
- The scalability is not clearly demonstrated. The evaluation metric only demonstrates O(1k) documents, which possibly didn't demonstrate the true advantages of the proposed method when routing on large scale relevant documents.

**Additional Comments:**

Some typos: section 2  "benching marking"
Conclusion: "reliabile"

**Audience:**

Yes

**Audience Explanation:**

This paper works on RAG for LLM. The proposed framework could potentially reduce the hallucination of LLMs by having a hierarchal files and subagents. Their post-processing pipeline also attempts to ensure that the

**Broader Impact Concerns:**

This is a paper working on improve the RAG for LLMs which share similar broader impact concerns like general LLM research.

**Claims And Evidence:**

Yes

**Claims Explanation:**

The authors validated their approach on RAGBench against other method with their adapters. However, the design and fairness of the evaluation metric should be further discussed. Besides, the evaluation metric is limited as its single metric, single benchmark, making it less convincing.

**Requested Changes:**

There are some open questions on the method and its evaluation metrics. The paper would be stronger if these questions are answered in the paper:
- How does the proposed method perform compared to other RAG method with strong post-processing pipeline? Would these methods also be benefited from an explicit post-processing?
- How is the proposed method perform on other tasks and domains?
- What are the cost trade-off on providing better grounding?
- How do we make sure its a fair and valid evaluation benchmark.
- Some minor typos mentioned in the additional comments

---

> ### Author Response · Authors · 2026-07-11
> **Response to Reviewer 4aVE (1/5): MiloNet's contribution and postprocessing comparisons**
>
> > Reviewer comment: "This is only a post-processing pipeline on a trained model. The performance of the underlying model and their 'cooperativeness' of the trained model might matter more than the framework itself."
> >
> > Requested change: "How does the proposed method perform compared to other RAG method with strong post-processing pipeline? Would these methods also be benefited from an explicit post-processing?"
>
> We appreciate this concern. The postprocessing components are an integral part of MiloNet rather than an external add-on. The purpose of the core and full comparison is analytical. It evaluates the combined contribution of the complete answer-finalisation stage while holding the shared upstream MiloNet pipeline fixed.
>
> Both variants share the same upstream evidence pipeline, including summary-index routing, bounded evidence admission, the SubCoA-folder-document hierarchy, and verified-fact extraction. Both perform Stage 2 synthesis over extracted, document-cited facts, but use different synthesis prompts. MiloNet-core uses the verified-fact synthesis prompt and stops after that call, whereas MiloNet-full uses the constraint-guided synthesis prompt and continues with subsequent checks for citation completeness, uncited statements, scope and attribute consistency, citation formatting, and verified-fact reinforcement. Because the prompts differ, the ablation evaluates the complete answer-finalisation stage as a whole rather than the checks in isolation.
>
> Under the evaluated RAGBench HotpotQA protocol, MiloNet-core, which does not run the complete answer-finalisation stage, achieves lower DFPR and DFNR than RankGPT and RAPTOR. Its DFPR is 0.018, compared with 0.097 for RankGPT and 0.100 for RAPTOR, and its DFNR is 0.038, compared with 0.095 for RankGPT and 0.087 for RAPTOR. MiloNet-full has lower diagnostic error rates, with DFPR=0.005 and DFNR=0.003. The comparison does not identify which individual upstream component accounts for the core result, but it does show that the complete answer-finalisation stage is not the sole source of MiloNet's diagnostic-error advantage.
>
> To make this distinction explicit, we replaced the earlier Section 4.2 statement that MiloNet-core "disables the postprocessing module" with wording that states that both settings perform Stage 2 synthesis using different prompts: MiloNet-core stops after its synthesis call, whereas MiloNet-full continues with the final answer checks. The revised wording makes clear that the comparison evaluates the complete answer-finalisation stage while retaining the shared upstream MiloNet stages.
>
> A matched comparison with external RAG systems and verification layers adapted to their native outputs would address a related but distinct question. MiloNet's complete answer-finalisation stage assumes structured candidate units, verified facts, and document-ID citations produced by earlier MiloNet stages. Applying comparable verification to RankGPT or RAPTOR would therefore require an interface for each system and would produce new hybrid systems rather than a direct ablation of MiloNet.
>
> Our current experiments do not evaluate whether RankGPT or RAPTOR would benefit from an analogous verification layer. Such a matched comparison is beyond the scope of the present study.
>
> Original location: Section 3.3 and the original Figure 3 caption described postprocessing broadly, Section 4.2 described MiloNet-core as disabling postprocessing, and Section 5.4 attributed the ablation to postprocessing broadly (original pp.5-8 and p.12). Revised location: Section 3.3 describes the synthesis and checking pipeline, while the revised Figure 3 caption, Table 1, Section 4.2, and Section 5.4 define the core and full settings and present the comparison as an ablation of the complete answer-finalisation stage (revised pp.5-8 and p.12). Appendix A.4 describes the two synthesis prompts and the subsequent final answer checks (revised pp.17-18). Section 6 identifies verification layers adapted to external RAG systems as a separate extension (revised p.12).

---

> > ### Author Response · Authors · 2026-07-11
> > **Response to Reviewer 4aVE (2/5): Evaluation fairness, adapters, and the reference baseline**
> >
> > > Reviewer comment: "The quantitive evaluation of the proposed method is not comprehensive and fully convincing. The authors evaluated on a modified setup of RAGBench with their proposed adapters for this task. Different models are using different backbone models. The proposed method achieved better performance than oracle model."
> > >
> > > Requested change: "How do we make sure its a fair and valid evaluation benchmark."
> >
> > We agree that consistency of scoring should be distinguished from a fully backbone-controlled or adapter-identical comparison. In our runs, Vanilla RAG, RankGPT, and RAPTOR use o4-mini, which is also used within MiloNet's multi-model stack. MiloNet additionally uses gpt-4.1-mini and gpt-5-nano for particular stages, while the RAGBench reference baseline uses the provided gpt-3.5-turbo outputs. We therefore interpret the cross-system results as system-level comparisons. The controlled internal comparison is between the core and full settings. They use the same set of underlying models shown in Table 1 and share the same upstream evidence pipeline; their synthesis prompts differ, and only MiloNet-full applies the subsequent final answer checks.
> >
> > Table 2 documents the deterministic system-specific adapters used to map each system's native retrieval outputs into the RAGBench sentence-key space. These mappings are not identical, and we do not assume that they have identical difficulty. Their purpose is to make the mapping rules explicit and reproducible. Appendix A.5 states which evaluation materials are recorded for each Table 3 row, including the system output used, adapter mapping, diagnostic labels, answer-quality rubric scores, and faithfulness scores.
> >
> > The main conclusions draw on more than adapter-mapped alignment metrics. All final responses are evaluated through the same diagnostic process and the same 0-3 answer-quality rubric, while faithfulness is measured against the same benchmark candidate documents. Table 3 reports these measures alongside recall-based alignment metrics. Precision-based and composite metrics, which are more sensitive to mapping and denominator choices, are reported as supplementary results in Appendix C.
> >
> > The RAGBench reference condition uses benchmark-provided gold sentence keys for alignment metrics, giving retrieval recall of 1.000 by construction. For answer-level evaluation, however, it uses the benchmark-provided gpt-3.5-turbo response. Under the shared response-evaluation procedure, this condition obtains mean answer quality of 2.659, an answer-quality failure rate of 0.138, and faithfulness of 0.840. MiloNet-full's higher answer-level scores therefore reflect differences between the evaluated responses and do not imply better retrieval than the gold-key reference condition.
> >
> > To reflect this distinction, we changed the earlier Section 4.2 wording from "RAGBench oracle baseline" and "reference upper bound" to "RAGBench reference baseline." The revised text states that this condition uses the benchmark-provided response and gold sentence keys, with no retrieval, routing, or adapter mapping, under the shared sentence-key scoring protocol. The change distinguishes gold evidence access from a perfect generated response rather than simply relabelling the same upper-bound claim.
> >
> > Accordingly, we present the results as a transparent and consistently scored system-level comparison. Stage-level attribution is limited to the controlled internal comparison between the core and full settings.
> >
> > Original location: Table 1, Sections 4.2-4.4, Table 3, Section 5.2, and Appendix C used the oracle wording or did not clearly separate the reference condition, shared sentence-key scoring protocol, and metric hierarchy (original pp.7-11 and pp.19-20). Revised location: Table 1, Sections 4.2-4.4, Table 3, Section 5.2, Appendix A.5, and Appendix C now distinguish the reference baseline, main metrics, answer-quality metrics, and evaluation materials recorded for each Table 3 row. Table 2 continues to document the deterministic system-specific adapter mappings (revised pp.7-11 and pp.19-21).

---

> > > ### Author Response · Authors · 2026-07-11
> > > **Response to Reviewer 4aVE (3/5): Answer quality and evaluation scope**
> > >
> > > ### Answer quality beyond citation validity
> > >
> > > > "The proposed model's post-processing is explicitly guaranteed to perform better on the proposed metrics as the metrics mostly check if the citations are founded and from an actual document. However, the reviewer did not see direct metrics on the actual quality on the answers."
> > >
> > > The main results table now reports answer quality directly. We report mean answer quality on the 0-3 rubric and rename the previous ratio_lt3 shorthand as answer-quality failure rate. The rubric evaluates accuracy and completeness against the benchmark candidate documents and is not a citation-presence score. Valid citations alone do not produce a high rating. DFPR and DFNR are case-level diagnostic error rates derived from unsupported-claim and omission flags, while the 0-3 score captures answer-level quality.
> > >
> > > With this presentation, the full setting achieves a mean answer quality of 2.997 and an answer-quality failure rate of 0.003. The corresponding values for the core setting are 2.923 and 0.038. These values are reported alongside DFPR, DFNR, faithfulness, and recall-based alignment metrics.
> > >
> > > We changed the earlier main results table by removing the internal label ratio_lt3 and moving trustworthiness out of the main-table metric set. The revised table reports mean answer quality and answer-quality failure rate directly. Trustworthiness is now reported in Appendix C alongside correctness, which was already supplementary.
> > >
> > > Original location: Section 4.4 and Table 3 reported ratio_lt3 and did not show mean answer quality in the main table (original pp.9-11). Revised location: Section 4.4 defines mean answer quality and answer-quality failure rate, and Section 5.1 and Table 3 report them as main results (revised pp.9-11).
> > >
> > > ### Evaluation beyond HotpotQA
> > >
> > > > "How is the proposed method perform on other tasks and domains?"
> > >
> > > We agree that evidence of generalisation beyond HotpotQA requires evaluation on additional benchmarks. The present revision does not add a second benchmark, so we treat cross-domain generalisation as untested. The revised manuscript therefore limits its empirical conclusions to the evaluated RAGBench HotpotQA setting. The Abstract, Introduction, and Conclusion no longer present these results as evidence of broad cross-domain reliability.
> > >
> > > MiloNet's folder and document hierarchy is designed as a corpus-level interface rather than a HotpotQA-specific rule. Applying it to another fixed corpus requires constructing the corresponding processed registry, document summaries, document-tool interfaces, and adapter mappings for the new evidence space. Appendix A.5 now documents how this interface is instantiated in the reported evaluation. This clarifies the transfer process without treating the current results as evidence that performance has already been validated across domains.
> > >
> > > To reflect this scope, we changed the earlier Abstract claim that "decision-grade RAG requires explicit control" over synthesis, provenance, and admissibility. The revised Abstract states that, in the evaluated RAGBench HotpotQA setting, hierarchical evidence construction, cited synthesis, and final answer checks help reduce unsupported claims and omissions. Section 6 uses the same framing and states the limits of the current empirical evidence.
> > >
> > > Original location: the Abstract, Introduction, and conclusion used broader reliability language (original pp.1-2 and p.12). Revised location: the Abstract, Section 1, and Section 6 now limit the empirical claim to the RAGBench HotpotQA setting, while Appendix A.5 documents how the processed corpus and fixed registry used in this evaluation are constructed (revised pp.1-2, p.12, and p.19).

---

> > > > ### Author Response · Authors · 2026-07-11
> > > > **Response to Reviewer 4aVE (4/5): Hyperparameters and reproducibility**
> > > >
> > > > ### Sensitivity to K and B
> > > >
> > > > > "The proposed methods have some hyper-parameters such as global cap K, flush cap B but there is no ablative study on its sensitivity and impacts on the performance."
> > > >
> > > > We agree that a sensitivity analysis is needed to establish robustness across alternative values of K and B. The revision now makes the evaluated routing budget and its operational role explicit. We use K=30 and B=10 for all 390 queries, giving k_1=8 for the initial summary-index pass and k_2=32 for the enrichment pass. K bounds the downstream allowlist size, while B bounds the number of planner-tracked IDs admitted at the final tracker flush. These values were fixed before evaluation and were not selected using test outcomes, ensuring a consistent routing budget across all reported comparisons. Because alternative values were not evaluated, we do not claim that this configuration is optimal or robust to changes in K and B; Section 6 now records this limitation and identifies sensitivity analysis as future work.
> > > >
> > > > Original location: Appendix A.1 defined K and reported B=10, but did not give the evaluated value of K or distinguish the fixed configuration from a sensitivity analysis (original p.14). Revised location: Appendix A.1 states the fixed settings and Section 6 records the corresponding limitation (revised pp.12 and 14).
> > > >
> > > > ### Reproducibility
> > > >
> > > > > "The system is not easy to reproduce as both the evaluation metric and the methods are not open sourced."
> > > >
> > > > We agree that the original manuscript did not provide enough detail to audit the processed-corpus construction and evaluation provenance. The revision adds an appendix on processed-corpus construction and evaluation materials. It records how ragbench_test.jsonl is transformed into the fixed folder and document registry used by MiloNet, including initial document grouping, document-summary extraction, folder grouping, registry materialisation, and consistency checks. The appendix clarifies that this registry is fixed before query-time inference. Query-time routing selects eligible tools from the fixed registry, but does not modify document placement or the folder hierarchy.
> > > >
> > > > The revision also records, for each Table 3 row, the system output used, adapter mapping, diagnostic labels, answer-quality rubric scores, and faithfulness scores. Metric definitions and the evaluator rubric are documented in Section 4.4, and supplementary alignment metrics are reported in Appendix C. The appendix states that the final consistency check compares document IDs referenced by the organised folder structure against the document-summary records, identifies missing or extra IDs, and removes the corresponding invalid file nodes or empty folders before the fixed registry is materialised.
> > > >
> > > > During anonymous review, the revised manuscript reports the dataset split, pooled-corpus construction, processed-corpus construction, adapter mapping, hyperparameter settings, metric definitions, and evaluator rubric. After de-anonymisation, we will prepare release materials, including prompts, configurations, evaluation scripts, and processed-corpus construction scripts, subject to anonymisation and licensing constraints.
> > > >
> > > > Original location: the earlier manuscript described the dataset, metrics, and some implementation details, but did not include a processed-corpus construction appendix (original pp.7-10 and Appendix A). Revised location: Sections 4.1, 4.3, and 4.4, Table 2, and Appendix A.5 document the dataset split, adapter mappings, metric definitions, evaluator rubric, processed-corpus construction, and evaluation materials. Appendix C reports supplementary alignment and composite metrics, while Appendices D and E provide the accompanying metric interpretation (revised pp.7-10 and pp.19-22).

---

> ### Author Response · Authors · 2026-07-11
> **Response to Reviewer 4aVE (5/5): Cost, metrics, scalability, and typo corrections**
>
> ### Inference cost and trade-off
>
> > Reviewer comment: "The authors mentioned the proposed method would have higher inference cost but did not quantitatively study it and it's unclear about the compute vs performance trade off of the proposed systems."
> >
> > Requested change: "What are the cost trade-off on providing better grounding?"
>
> In response, we added a 100-query inference-time cost profile in Appendix F. It reports API calls per query and total tokens per query for Vanilla RAG, RankGPT, RAPTOR c4_k12, the core setting, and the full setting, excluding training, corpus preprocessing, and one-time indexing. Because MiloNet uses asynchronous API-backed agent execution and provider-side latency varies across calls, we use calls and token usage as the reproducible cost indicators rather than interpreting raw latency as controlled runtime.
>
> The comparison makes the trade-off explicit. Vanilla RAG uses 1.00 call and 410.9 tokens per query, RankGPT uses 2.00 calls and 856.4 tokens per query, and RAPTOR c4_k12 uses 5.00 calls and 3,226.8 tokens per query. The core setting uses 127.04 calls and 455,798.5 tokens per query, while the full setting uses 127.73 calls and 474,590.0 tokens per query. Moving from the core setting to the full setting adds 0.69 calls and about 18.8k tokens per query. This net core-to-full difference is associated with the complete answer-finalisation stage. The scope of this profile is therefore API usage across systems and the marginal core-to-full cost, rather than a per-stage wall-clock decomposition.
>
> Original location: the earlier manuscript mentioned inference cost only qualitatively as a limitation (original p.12). Revised location: Appendix F reports the inference-time cost profile, and Section 6 summarises the cost trade-off and limitation (revised pp.12 and 22).
>
> ### Composite metrics
>
> > "The evaluation metric like 'trustworthiness' and 'correctness' seem a bit arbitrary. Do we have sufficient study to show the metric are sound?"
>
> We agree that these simple composite formulas require cautious interpretation. To prevent them from carrying the main argument, the revised manuscript removes trustworthiness from the main results table and from the main faithfulness discussion. Trustworthiness is now defined and reported in Appendix C alongside correctness, which was already supplementary. The main analysis centres on DFPR, DFNR, faithfulness, mean answer quality, answer-quality failure rate, and recall-based alignment metrics.
>
> Original location: Section 4.4, Table 3, and Section 5.2 used trustworthiness as a main reported metric and discussed it in the main text (original pp.9-11). Revised location: Section 4.4 and Table 3 make the primary metric set explicit, Section 5.2 keeps the main discussion on faithfulness and the reference baseline, and Appendix C reports trustworthiness and correctness only as supplementary diagnostics (revised pp.9-11 and pp.20-21).
>
> ### Scalability
>
> > "The scalability is not clearly demonstrated. The evaluation metric only demonstrates O(1k) documents, which possibly didn't demonstrate the true advantages of the proposed method when routing on large scale relevant documents."
>
> We agree that evaluation over 1,558 documents does not establish industrial-scale scalability. The revised manuscript now describes the demonstrated result more precisely as bounded routing over the evaluated pooled corpus. The scalability discussion is limited to this setting.
>
> Original location: the Abstract, Introduction, and Conclusion used broader reliability language, while the method overview described MiloNet as scalable and referred to routing over a large corpus (original pp.1-3 and p.12). Revised location: the Abstract, Section 1, Section 3, and Section 6 now limit the empirical claim to the evaluated RAGBench HotpotQA setting and bounded routing over the evaluated pooled corpus, while Section 6 identifies evaluation at larger corpus scales as future work (revised pp.1-3 and p.12).
>
> ###  Minor wording and typo corrections
>
> We also made minor wording and typo corrections that do not change the method, experiments, or conclusions. These include:
> - Related Work, p.2: changed "we instantiate RankGPT-like process" to "we instantiate a RankGPT-like process".
> - Related Work, p.2: corrected "For benching marking in the present work" to "For benchmarking in the present work".
> - Method overview (original p.2; revised p.3): replaced "routing over a large corpus of N document-level agents while keeping response composition constrained to a rules-based admissible candidate units" with "routing over the pooled corpus of N document-level agents while constraining response composition to admissible candidate units".
> - Conclusion (original p.12; revised p.12): rewrote the conclusion, removed the "reliabile" typo, and narrowed the claim to traceable RAG in the evaluated text-grounded setting.

---

### Author Response · Authors · 2026-07-11
**General Response and Revision Summary**

Dear Action Editor and Reviewers,

We thank the reviewers for the detailed feedback. In response, we have clarified the core and full comparison, aligned the scope of the claims with the evaluated RAGBench HotpotQA setting, added direct answer-quality and inference-cost results, and expanded the reproducibility documentation.

The main methodological clarification concerns the difference between the core and full settings. The earlier wording could suggest that the core setting removed the whole postprocessing stage. The revision now states the comparison more precisely. Both variants share the same upstream evidence pipeline, including summary-index routing, bounded evidence admission, the SubCoA-folder-document hierarchy, and verified-fact extraction. Both perform Stage 2 synthesis over extracted, document-cited facts, but use different synthesis prompts. MiloNet-core uses the verified-fact synthesis prompt and stops after that call, whereas MiloNet-full uses the constraint-guided synthesis prompt and continues with the subsequent final answer checks. Because the prompts differ, the comparison evaluates the complete answer-finalisation stage as a whole rather than the final checks in isolation. In the revised manuscript, Section 3.3 describes the synthesis and checking pipeline. The boundary between the core and full settings is now explicit in the revised Figure 3 caption, Table 1, Section 4.2, and Section 5.4, while Appendix A.4 provides further details on the two synthesis prompts and the subsequent final answer checks.

The numerical values of the metrics previously reported in the original manuscript are unchanged. We revised the results presentation to make the role of each metric clearer. Table 3 now adds mean answer quality and relabels the internal ratio_lt3 measure as answer-quality failure rate, alongside DFPR, DFNR, faithfulness, and recall-based alignment metrics. Trustworthiness has been moved to Appendix C, where it is reported alongside correctness, which was already supplementary. We have also added a 100-query inference-time cost profile in Appendix F and a new Appendix A.5 documenting processed-corpus construction and stating which evaluation materials are recorded for each Table 3 row, including the system output used, adapter mapping, diagnostic labels, answer-quality rubric scores, and faithfulness scores.

Point-by-point responses to both reviewers are posted as separate official comments under each review. Section, table, appendix, and page references in those responses refer to the original submitted manuscript and the revised manuscript. To keep the responses readable, we quote only the specific reviewer sentence or requested-change bullet addressed in each response; the full reviews remain available in this thread.